# Mechanistic insight into bacterial entrapment by septin cage reconstitution

Damián Lobato-Márquez ⓘ [1]✉, Jingwei Xu ⓘ [2], Gizem Özbaykal Güler[1], Adaobi Ojiakor[1], Martin Pilhofer[2] & Serge Mostowy ⓘ [1]✉

Septins are cytoskeletal proteins that assemble into hetero-oligomeric complexes and sense micron-scale membrane curvature. During infection with *Shigella flexneri*, an invasive enteropathogen, septins restrict actin tail formation by entrapping bacteria in cage-like structures. Here, we reconstitute septin cages in vitro using purified recombinant septin complexes (SEPT2-SEPT6-SEPT7), and study how these recognize bacterial cells and assemble on their surface. We show that septin complexes recognize the pole of growing *Shigella* cells. An amphipathic helix domain in human SEPT6 enables septins to sense positively curved membranes and entrap bacterial cells. *Shigella* strains lacking lipopolysaccharide components are more efficiently entrapped in septin cages. Finally, cryo-electron tomography of in vitro cages reveals how septins assemble as filaments on the bacterial cell surface.

[1] Department of Infection Biology, London School of Hygiene and Tropical Medicine, London, UK. [2] Department of Biology, Institute of Molecular Biology and Biophysics, Eidgenössische Technische Hochschule Zürich, Zürich, Switzerland. ✉email: damian.marquez@lshtm.ac.uk; serge.mostowy@lshtm.ac.uk

**S**higella flexneri is an important human-adapted bacterial pathogen that causes ~160 million illness episodes per year[1]. S. flexneri infects the small intestine and invades colonic epithelial cells, causing inflammation and tissue destruction[2]. Shortly after invasion, S. flexneri breaks the phagocytic vacuole and replicates in the cytosol of infected cells where it can evade cell-autonomous immune responses[3]. Successful S. flexneri infection depends on the plasmid-encoded type 3 secretion system (T3SS) and IcsA[4]. Localized on the outer membrane at one pole of the bacterial cell[5], IcsA recruits host proteins N-WASP and Arp2/3 complex to polymerize actin tails and disseminate from cell-to-cell[6,7]. During infection, lipopolysaccharide (LPS) protects bacteria against antimicrobial insults (e.g., attack by complement)[8] and helps to localize IcsA at the bacterial cell pole for efficient actin tail formation[9].

To counteract S. flexneri infection, host cells employ cell-autonomous immune responses, including antibacterial autophagy[10], coating by IFNγ-inducible guanylate binding proteins (GBPs)[11–13], and septin cage entrapment[14–16]. Both GBPs and septins can block actin tail formation, but the relationship between these two cell-autonomous immune responses is only starting to emerge[17], and new work has shown that GBPs bind and disrupt LPS, delocalizing IcsA from the bacterial cell pole[18]. Septins are cytoskeletal proteins that play key roles in cell-autonomous immunity by sensing micron-scale membrane curvature[19–21]. Human cells encode 13 septin paralogs that are classified into 4 separate groups according to amino acid sequence identity[22,23]. Subunits from one of each of the 4 septin groups form hetero-oligomeric complexes that assemble into non-polar filaments and higher-order structures such as bundles, rings, lattices, and gauzes[22,23]. During S. flexneri infection, actin polymerizing bacteria are recognized by septins and entrapped in cage-like structures that restrict motility and target bacteria to destruction by autophagy[16,24]. Despite recent insights, the mechanisms underlying recognition of bacteria for septin cage entrapment are mostly unknown.

Cell-free systems are powerful platforms to decipher molecular mechanisms of processes happening in biochemically complex environments, including the host cell[25]. The discovery of Arp2/3-mediated actin polymerization using bacteria and cell-free extracts highlights the great potential of 'bottom-up' cellular microbiology[26,27].

In this work, using an in vitro reconstitution system based on purified recombinant septin complexes, we discover how septins recognize S. flexneri for cage entrapment and also how bacteria can avoid this process.

## Results

### In vitro reconstitution of bacterial septin cages.
Whether septin recognition of S. flexneri is due to direct interaction with the bacterial surface or requires additional host cell factors was unknown. To test this, we designed a cell-free system based on purified recombinant septin complexes. We selected SEPT2–SEPT6–SEPT7 because this septin hetero-oligomer is well characterized[28,29] and these three septins assemble into cages in human epithelial cells upon S. flexneri infection[14–16]. To visualize septin complexes using fluorescence microscopy, we fused a monomeric superfolder green fluorescent protein (msGFP) to the N-terminus of human SEPT6 (Fig. 1a, Supplementary Fig. 1). We then incubated the purified SEPT2–msGFP-SEPT6–SEPT7 complex with S. flexneri str. M90T (Fig. 1a). In the absence of additional host cell factors, Airyscan confocal microscopy showed filamentous septins wrapping around bacterial cells, suggesting that reconstituted septin cages closely resemble those formed during infection of human epithelial cells (Fig. 1b). However, septin-bacteria interactions were infrequent

under these experimental conditions (<1 recruitment event per 20,000 bacteria). Previous work using infected tissue culture cells suggested that bacterial cell growth can promote septin cage entrapment[14]. Considering that our initial in vitro reconstitution assay did not provide nutrients required for bacterial cell growth, we adapted our conditions by using an M9-based minimal medium to (i) promote bacterial growth and replication (in this case, S. flexneri replicates $1.6 \pm 0.1$-fold over 2 h) and (ii) permit septin assembly into filaments. Under these growth conditions, septins are recruited to $61.3 \pm 3.7\%$ of growing bacterial cells, and are often bound to one bacterial cell pole (Fig. 1c), mimicking the initial steps of septin caging in human epithelial cells[14]. Consistent with bacterial growth-promoting septin recruitment, bacterial sedimentation assays (where the amount of septins bound to bacteria was quantified by immunoblotting, Fig. 1a) showed a $52.7 \pm 9.3$-fold increase in the amount of SEPT7 bound to growing S. flexneri cells (as compared to non-growing S. flexneri cells) (Fig. 1d, e). Together, these data demonstrate that septins directly bind the surface of S. flexneri, and bacterial growth is essential for septin recruitment in vitro.

Other bacterial species, including mycobacteria, are recognized by septins for cage entrapment[16]. To test if our in vitro reconstitution system can be applied to different bacterial species, we used Mycobacterium smegmatis and Mycobacterium marinum. Both mycobacterial species are highly recognized by septins in vitro ($67.8 \pm 8.1\%$ M. smegmatis and $65.2 \pm 8.8\%$ M. marinum) and in most cases septins assemble into cage-like structures that cover the entire surface of mycobacterial cells (Fig. 1f and Supplementary Fig. 2a). This situation contrasts with S. flexneri, where septin binding is dependent on bacterial growth and mainly occurs at one pole of the bacterial cell.

Septins assemble into non-polar filaments and higher-order structures such as bundles, rings, lattices, and gauzes[22,23]. Previously, complexity of the host cell cytosol combined with limitations of resolution have prevented the visualization of septin assembly on the bacterial surface. To overcome these limitations, we combined our in vitro reconstitution system with cryo-Electron Tomography (cryoET) (Fig. 1a). Given the efficient binding of septins to M. smegmatis in vitro, we selected this bacterial species for cryoET studies. Remarkably, cryoET showed that septins assemble as irregularly distributed filaments wrapping around the surface of M. smegmatis, extending parallel to the short axis of the bacterial cell (Fig. 1g and Supplementary Fig. 2b). Septin filaments around M. smegmatis are spaced $15.9 \pm 0.3$ nm from the mycobacterial outer membrane (Fig. 1h), a distance that may represent the capsular layer of mycomembrane rich in mycobacterial lipids[30,31]. To confirm that structures we observe by cryoET on the surface of bacteria are septin filaments, we imaged M. smegmatis not incubated with septins in vitro; in this case, we did not observe any structures bound to the bacterial surface (Supplementary Fig. 2c). Together, our in vitro reconstitution system enables the visualization of septin filaments on bacterial surfaces at the nanometer scale.

### IcsA promotes the recruitment of septins to one pole of S. flexneri.
Recent work has shown that bacteria void of cardiolipin are less efficiently recognized by septins[14]. As proof of concept, we tested a S. flexneri mutant lacking the synthesis pathway for cardiolipin (ΔCL) in our in vitro reconstitution system. In agreement with the ability of septins to recognize cardiolipin during infection, septins bound S. flexneri ΔCL $3.3 \pm 0.9$-fold less than S. flexneri WT in vitro (Supplementary Fig. 3a, b). We confirmed these data with bacterial sedimentation assays, where a significant reduction ($1.8 \pm 0.3$-fold) in the amount of bound SEPT7 to S. flexneri ΔCL (as compared to S. flexneri WT) is observed (Supplementary Fig. 3c, d).

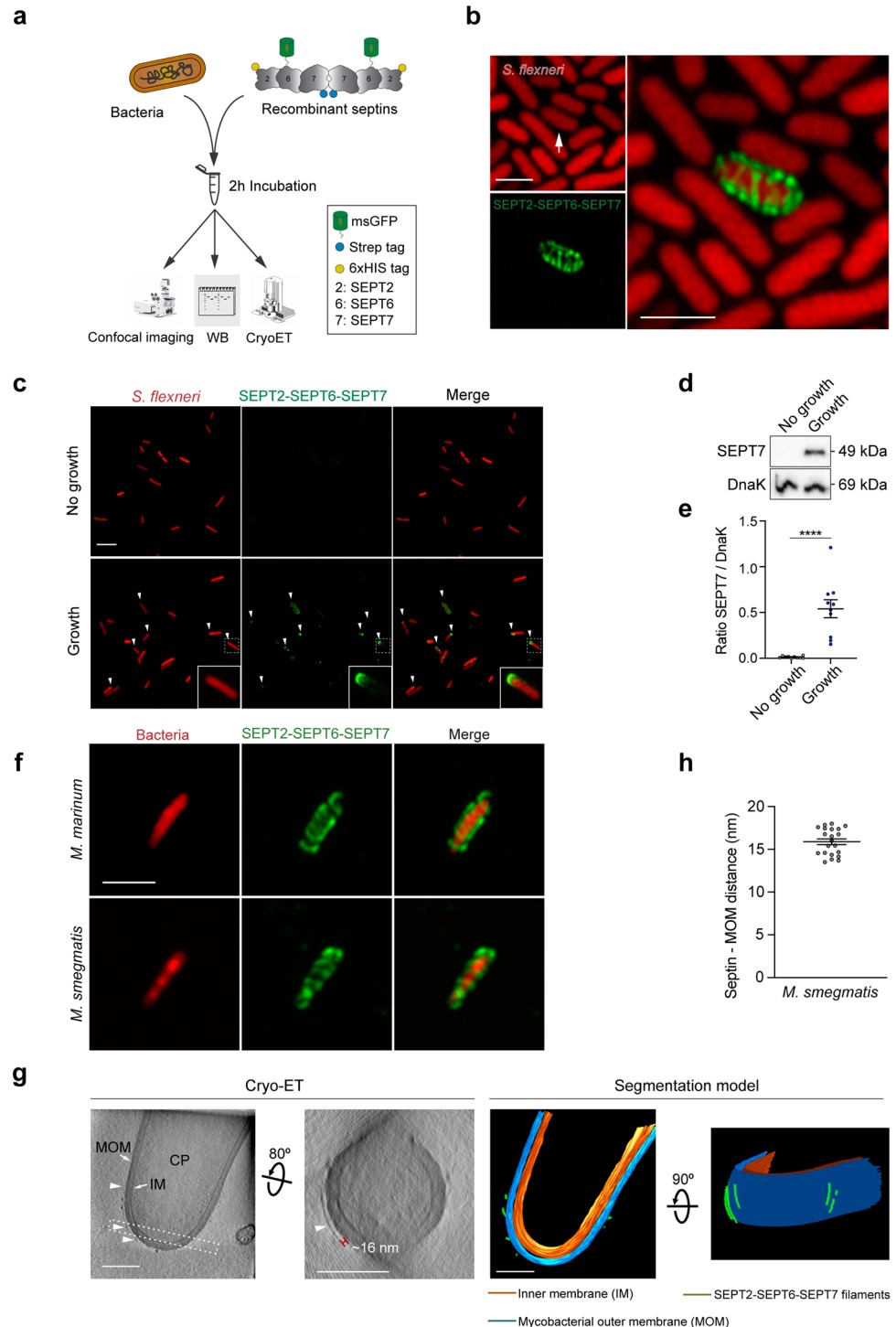

**Fig. 1 Bacterial growth is essential for *S. flexneri* recognition by septins. a** Pipeline followed for the analysis of in vitro reconstituted septin cages. CryoET, cryo-electron tomography; WB, western blotting. **b** Representative Airyscan confocal image of a *S. flexneri* 5a str. M90T septin cage reconstituted in vitro. White arrow, bacterial cell entrapped in a septin cage. Scale bar, 2 μm. **c** Airyscan confocal images comparing septin binding in the absence (top) or presence (bottom) of bacterial growth. White arrowheads, bacteria binding polar septin complexes. This experiment was performed 10 independent times. Scale bar, 5 μm. **d, e** Bacterial sedimentation assays of samples from panel (**c**). A representative blot is shown (**d**). **e** Graphs represent the mean ± SEM of the ratio SEPT7/DnaK (used as loading control) from 10 independent blots. ****$p < 0.0001$ by two-tailed Student's $t$-test. **f** Representative Airyscan confocal images of *M. marinum* (top) and *M. smegmatis* (bottom) septin cages reconstituted in vitro. This experiment was performed 3 independent times. Scale bar, 2 μm. **g** CryoET images of a *M. smegmatis* septin cage reconstituted in vitro (left, accession No. #EMD-12562). The corresponding segmentation models are shown in the right panels. Image shown corresponds to a slice of 10.8-nm thickness. CP, cytoplasm; MOM, mycobacterial outer membrane; IM, inner membrane. Arrowheads, septin filaments. Dashed rectangle, bacterial cell area selected to show the septin filament. Scale bars, 200 nm. **h** Measured distance between septin filaments and the *M. smegmatis* outer membrane. Data correspond to mean ± SEM from $n = 21$ filaments from 3 independent tomograms.

Although cardiolipin is important for S. flexneri septin cage entrapment, the percentage of septin caged bacteria lacking cardiolipin is only reduced ~1.4-fold during infection (as compared to S. flexneri presenting cardiolipin)[14]. These data argue for additional septin targets on the bacterial cell surface. Secreted at one pole, IcsA diffuses along the bacterial membrane forming a gradient to the other pole[32]. Considering this, we hypothesized that recognition of the S. flexneri pole by septins depends on IcsA and quantified septin recruitment to S. flexneri WT or S. flexneri ΔicsA in vitro. Strikingly, the percentage of S. flexneri ΔicsA binding septins is significantly reduced (7.6 ± 1.8-fold) as compared to S. flexneri WT (Fig. 2a, b). The septin binding defect of S. flexneri ΔicsA is recovered when icsA is expressed in trans from a plasmid (Fig. 2a, b). Escherichia coli is taxonomically indistinguishable from S. flexneri, but does not carry the virulence plasmid that encodes icsA and is not recognized by septins in vitro (Fig. 2c, d). However, when icsA is introduced in E. coli and expressed in trans from a plasmid, the percentage of bacteria recruiting septins is significantly increased (5.9 ± 0.6-fold) (Fig. 2c, d). These data suggest that septin recognition of the bacterial surface requires IcsA, and likely explain why S. flexneri ΔicsA is not recognized by septins during infection of HeLa cells[16]. To further test this, we employed an mCherry-tagged IcsA[33] and tracked the relative position of IcsA in relation to septins in vitro. In support of our hypothesis, 92.9 ± 2.7% of bacteria recruiting septins also have mCherry-tagged IcsA localized to the same bacterial pole as septins (Fig. 2e). These findings highlight a new role for IcsA in promoting the recognition of S. flexneri by septins for cage entrapment.

We also tested the role of MxiD (an essential component of the T3SS) and IcsB (a T3SS effector correlated with avoidance of septin caging during infection[16]) in our in vitro reconstitution system. The T3SS is crucial for host cell invasion and escape of S. flexneri to the cytosol[2,34]; as a result, S. flexneri mutants lacking the T3SS are not entrapped in septin cages during infection[16]. Inside HeLa cells, IcsB is well known to block autophagy[10] and septin caging[16]. Here, we tested whether the T3SS and IcsB may have a direct role in septin cage assembly, and show that neither the T3SS nor IcsB influence septin-bacteria interactions under our experimental conditions in vitro (Fig. 2b). However, we cannot rule out that specific T3SS effectors may interfere with S. flexneri septin cage assembly in vivo by targeting additional host factors not present in our in vitro system.

Considering that GBPs delocalize IcsA from the bacterial cell pole[18], we sought to determine if GBPs promote or inhibit septin cage entrapment. To test this, we infected HeLa cells (treated or not with IFNγ, Supplementary Fig. 4a, b) with S. flexneri and quantified the percentage of septin caged and/or GBP-decorated bacteria. In agreement with previous work[11–13], we observed a significant increase in the number of GBP1 positive S. flexneri upon IFNγ stimulation (21.9 ± 1.8%) as compared to non-stimulated host cells (0.5 ± 0.3%). In contrast, the percentage of septin caged S. flexneri upon IFNγ stimulation (9.9 ± 0.7%) was slightly decreased as compared to non-stimulated host cells (12.9 ± 0.9%), and only 1.2 ± 0.5 % of bacteria were positive for both GBP1 and septin cages (Supplementary Fig. 4c, d). Together, these results suggest that septins are not reliant upon IFNγ stimulation to restrict S. flexneri actin tail formation, and that septins and GBPs play complementary roles in cell-autonomous immunity.

### The amphipathic helix domain of SEPT6 senses membrane curvature and is important for cage entrapment. In vitro work using purified septins has shown that yeast septin Cdc12 encodes an amphipathic helix (AH) domain required for sensing of micron-

scale positive membrane curvature[35]. In silico screening revealed a putative AH domain encoded in human SEPT6[35] (Fig. 3a), therefore we hypothesized that the SEPT6 AH domain may enable human septins to sense micron-scale positive membrane curvature. We first tested if SEPT6 contains a bona fide AH and engineered HeLa cells producing recombinant msGFP–SEPT6WT or SEPT6 lacking the AH (SEPT6ΔAH) (Fig. 3b, c). Recombinant msGFP–SEPT6WT and msGFP–SEPT6ΔAH both co-localized with endogenous SEPT7, confirming they could assemble with endogenous septins and form hetero-oligomers (Fig. 3d). However, Airyscan confocal microscopy clearly demonstrated that msGFP–SEPT6WT, but not msGFP–SEPT6ΔAH, associated with positive curvature at the plasma membrane (Fig. 3d). These data show that similar to yeast septin Cdc12, human SEPT6 employs an AH domain to sense micron-scale membrane curvature, and in the absence of the SEPT6 AH domain septin complexes fail to localize to sites of membrane curvature.

Micron-scale membrane curvature is important for septin cage entrapment[14]. We sought to determine if the SEPT6 AH domain is involved in bacterial sensing. To test this, we purified septin complexes SEPT2–msGFP-SEPT6WT–SEPT7 and SEPT2–msGFP–SEPT6ΔAH–SEPT7 and mixed them with S. flexneri in vitro (Fig. 3e). Although we did not observe significant differences in the percentage of bacteria recruiting septins in vitro in the presence of SEPT6 WT or SEPT6ΔAH (Fig. 3f and Supplementary Fig. 5), we did observe a significant decrease (1.3 ± 0.1-fold) in the total amount of SEPT7 bound to bacteria in samples containing purified SEPT6ΔAH (Fig. 3g, h). To test the role of the SEPT6 AH domain during infection, we infected HeLa cells producing msGFP–SEPT6WT or msGFP–SEPT6ΔAH with S. flexneri and quantified the percentage of septin caged bacteria. In this case, we observed a significant reduction (2.9 ± 0.7-fold SEPT7 cages and 3.0 ± 0.7-fold msGFP-SEPT6ΔAH cages) in the percentage of septin caged bacteria in HeLa cells producing msGFP–SEPT6ΔAH (Fig. 3i). Together, these data show that the AH domain of SEPT6 is important for septin cage entrapment of S. flexneri cells during infection.

### Bacterial lipopolysaccharide protects S. flexneri from septin cage entrapment. Previous work has shown that the polysaccharide component of LPS (i.e., O-antigen) can mask IcsA on the outer membrane of S. flexneri and help localize IcsA at one bacterial pole for efficient actin tail motility[9,36,37]. We therefore hypothesized that S. flexneri may use LPS to mask IcsA on the bacterial cell surface and prevent septin cage entrapment. To test this, we deleted rfaC (also known as waaC) from S. flexneri; rfaC encodes an heptosyltransferase that links the first two sugars of the inner LPS core to lipidA during LPS synthesis (Supplementary Fig. 6a). In agreement with previous work[9,18,38], the lack of O-antigen, outer and inner core of LPS destabilized the S. flexneri outer membrane, provoking an increased sensitivity to the outer membrane-targeting detergent SDS and a significant reduction in actin tail formation (Supplementary Fig. 6b–d). To further investigate the protective role of LPS on IcsA, we tested septin recruitment to S. flexneri ΔrfaC using our in vitro reconstitution system. In support of LPS masking IcsA, 82.8 ± 5.1% of S. flexneri ΔrfaC cells are recognized by septins as compared to 54.7 ± 6.2% of S. flexneri WT (Fig. 4a, b). Although septins are mostly bound to one pole of S. flexneri WT, S. flexneri ΔrfaC are fully entrapped in septin cages (Fig. 4a). Septin cages obtained in vitro with S. flexneri ΔrfaC closely resemble septin cages observed during infection of human epithelial cells with S. flexneri ΔrfaC (Supplementary Fig. 7). To test the protective role of LPS against septin caging, we infected HeLa cells with S. flexneri WT or ΔrfaC and quantified the percentage of bacteria entrapped in SEPT7 cages. Here, S. flexneri ΔrfaC is

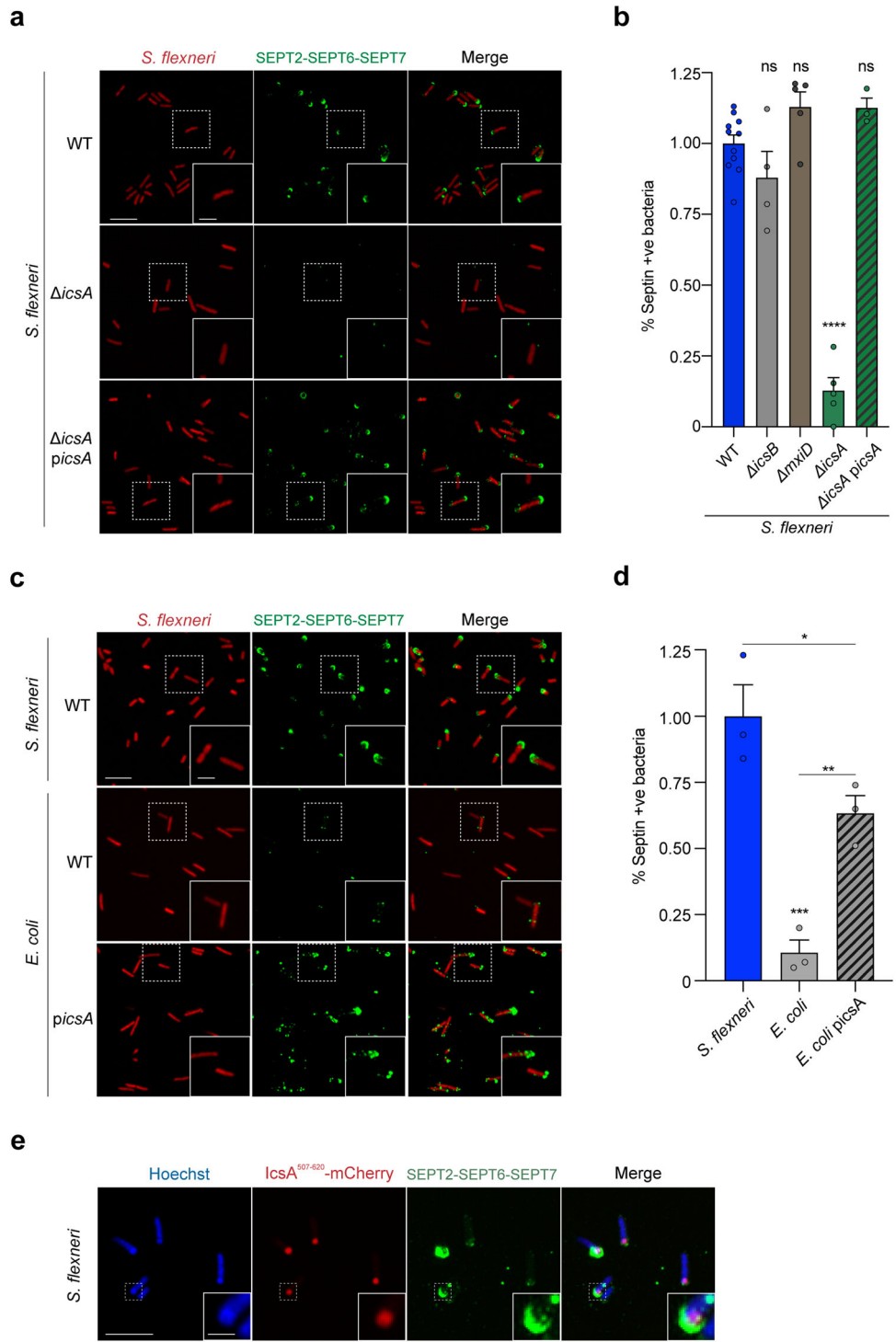

**Fig. 2 Septins recognize IcsA on the outer membrane of *S. flexneri*. a** Airyscan confocal images showing the binding of septins in vitro to *S. flexneri* WT (top), Δ*icsA* (middle), or Δ*icsA* complemented with *icsA* from a plasmid (bottom). Scale bar, 5 μm (inset, 2 μm). **b** Percentage of different bacterial strains recruiting septins in vitro normalized to *S. flexneri* WT mean value. Data represent mean ± SEM from $n = 2643$ (WT), $n = 1262$ (Δ*icsA*), $n = 2358$ (Δ*icsB*), $n = 1631$ (Δ*mxiD*), and $n = 880$ (Δ*icsA* + picsA) *S. flexneri* cells distributed in at least 3 independent experiments. ns, non-significant with $p = 0.270$ (Δ*icsB*), $p = 0.230$ (Δ*mxiD*), and $p = 0.254$ (Δ*icsA* + picsA), ****$p < 0.0001$ by one-way ANOVA and Dunnett's post-test. **c** Airyscan confocal images showing the binding of septins in vitro to *S. flexneri* WT (top), *E. coli* str. BL21 (middle), or *E. coli* str. BL21 complemented with *icsA* from a plasmid (bottom). Scale bar, 5 μm (inset, 2 μm). **d** Percentage of bacteria recruiting septins in vitro normalized to *S. flexneri* WT mean value. Data represent mean ± SEM from $n = 1073$ (*S. flexneri* WT), $n = 669$ (*E. coli*), and $n = 1132$ (*E. coli* + picsA) distributed in at least 3 independent experiments. *$p = 0.0447$, **$p = 0.0095$, ***$p = 0.0006$ by one-way ANOVA and Tukey's post-test. **e** Airyscan confocal images showing co-localization of septins and IcsA$^{507-620}$-mCherry at the pole of *S. flexneri* str. M90T. This experiment was performed 3 independent times. Scale bar, 5 μm (inset, 1 μm).

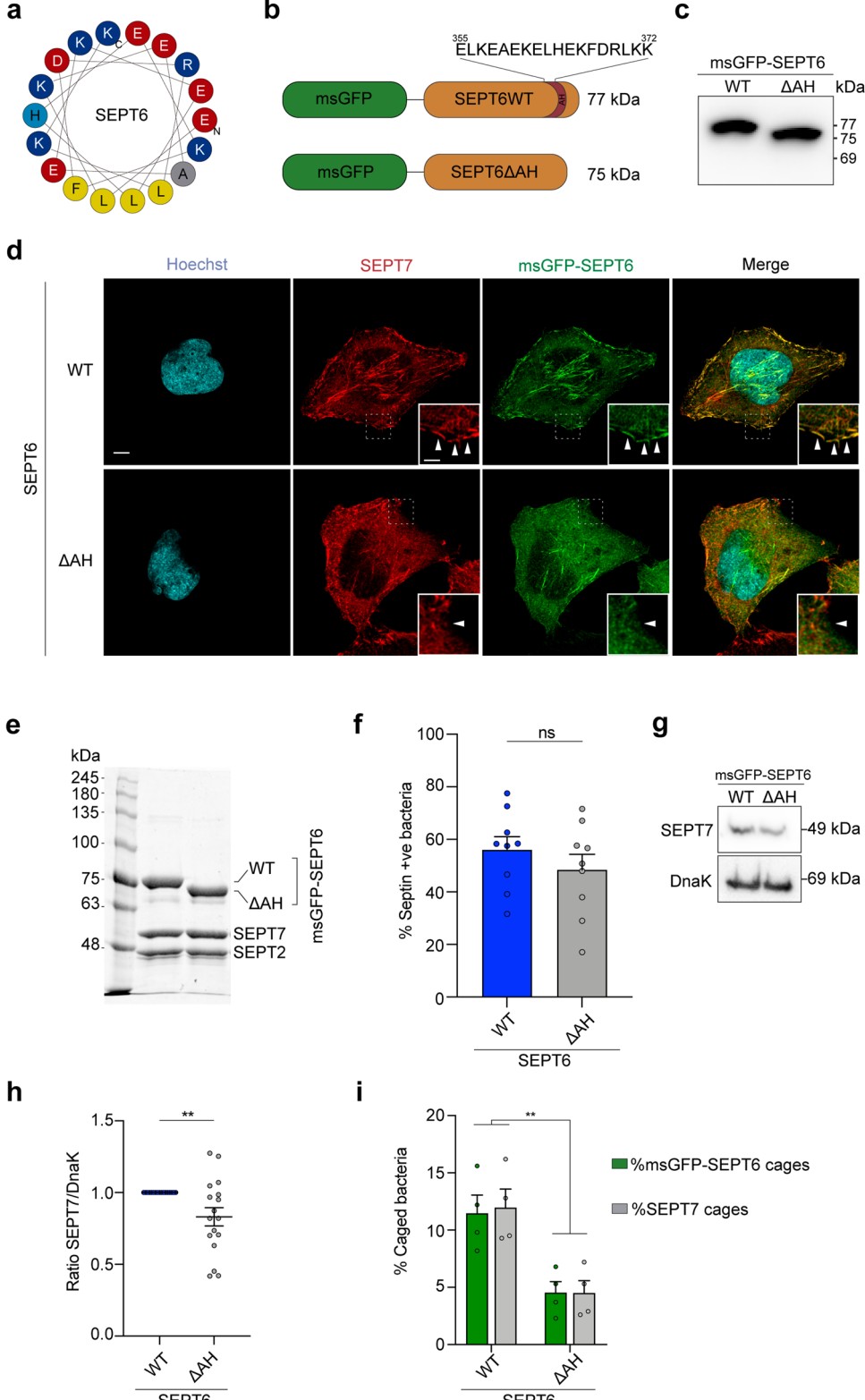

entrapped in SEPT7 cages significantly more often (2.1 ± 0.3-fold) than *S. flexneri* WT (Fig. 4c, d).

Bacteria with severe truncations of the LPS (deep-rough mutants) are known to have altered lipid and protein composition, with pleiotropic effects on bacterial outer membrane[39]. To discard the possibility that altered outer membrane of *S. flexneri* Δ*rfaC* is responsible for increased septin recruitment, we removed different LPS components. We engineered *S. flexneri*

mutants lacking O-antigen (Δ*rfaL*) or O-antigen and outer core components of the LPS (Δ*galU*) (Supplementary Fig. 8a), and tested for septin recruitment in our in vitro reconstitution system. Consistent with a role for bacterial LPS in preventing septin cage entrapment, *S. flexneri* Δ*rfaL* and Δ*galU* are recognized by septins as efficiently as *S. flexneri* Δ*rfaC* in vitro (Supplementary Fig. 8b). Septin recruitment to *S. flexneri* Δ*galU* is similar to that of *S. flexneri* Δ*rfaC*, i.e., the entire bacterial

**Fig. 3 A C-terminal SEPT6 amphipathic helix domain is important for membrane curvature sensing and cage entrapment. a** Wheel diagram representing SEPT6 AH domain was generated using Heliquest[59]. Adapted from ref. [35]. AH features: Hydrophobic moment ($\mu H$), 0.567, Net charge ($z$), 0. Blue, positively charged residues; light blue, uncharged residues; red, negatively charged residues; yellow, hydrophobic residues; grey, alanine. **b** Design of recombinant msGFP–SEPT6 constructs. Amino acid positions of the SEPT6 AH are indicated. **c** Representative western blot (anti-GFP) showing the correct production and molecular size of msGFP–SEPT6WT and msGFP–SEPT6ΔAH in HeLa cells. This blot was performed 3 independent times. **d** Representative Airyscan confocal images showing the distribution of SEPT6 and SEPT7 in HeLa cells stably producing msGFP–SEPT6 WT (top) or ΔAH (bottom). This experiment was performed 3 independent times. Arrowheads, positive membrane curvature recruiting septins. Scale bar, 5 μm (inset, 2 μm). **e** Coomassie blue staining of purified septin complexes SEPT2–msGFP–SEPT6–SEPT7 (left) or SEPT2–msGFP–SEPT6ΔAH–SEPT7 (right). Representative image from 2 independent experiments. **f** Percentage of bacteria recruiting septins in vitro using purified complexes from (**e**). Data represent mean ± SEM from $n = 3758$ (SEPT6 WT) and $n = 3039$ (SEPT6ΔAH) S. flexneri cells distributed in 9 independent experiments. ns, $p = 0.335$ by two-tailed Student's $t$-test. **g** Bacterial sedimentation assays of samples from (**f**). A representative blot is shown (**g**). **h** Graphs represent the ratio of SEPT7/DnaK (used as loading control) normalized to SEPT6WT (including normalized values from Fig. 1e). Data represent the mean ± SEM from 17 blots. **\*\***$p = 0.039$ by two-tailed Mann–Whitney's test. **i** Percentage of septin caged bacteria in msGFP–SEPT6WT- or msGFP–SEPT6ΔAH-producing HeLa cells infected for 3 h 40 min. Data represent the mean ± SEM from $n = 1374$ (msGFP–SEPT6WT) and $n = 1322$ (msGFP–SEPT6ΔAH) S. flexneri cells distributed in 4 independent experiments. **\*\***$p < 0.01$ by two-way ANOVA and Sidak's post-test.

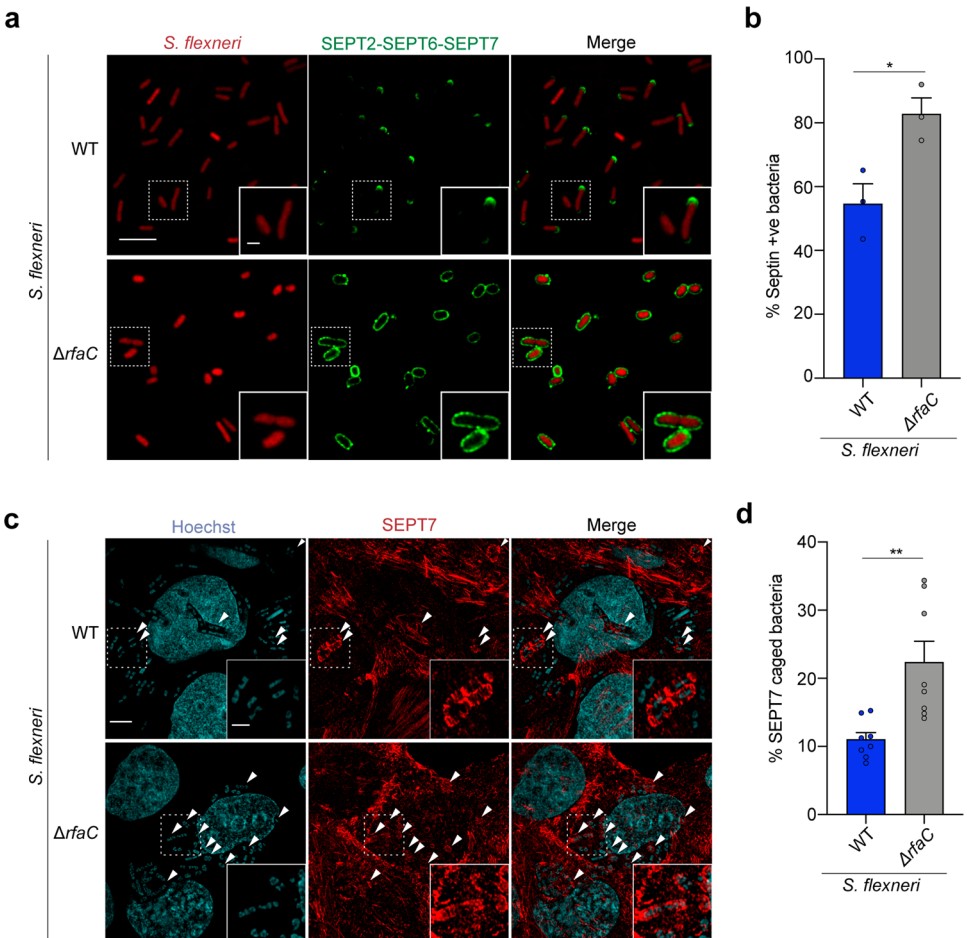

**Fig. 4 Lipopolysaccharide protects S. flexneri from septin cage entrapment. a** Representative Airyscan confocal images showing the binding of septins in vitro to S. flexneri (WT, top) or ΔrfaC (O-antigen⁻, bottom). Scale bar, 5 μm (inset, 1 μm). **b** Percentage of bacteria recruiting septins in vitro. Quantifications represent mean ± SEM from $n = 895$ (WT), $n = 689$ (ΔrfaC) S. flexneri cells distributed in 3 independent experiments. **\***$p = 0.0249$ by two-tailed Student's $t$-test. **c** Airyscan confocal images of HeLa cells infected for 3 h 40 min with S. flexneri str. afaE (WT, top) or S. flexneri ΔrfaC (bottom) and stained for SEPT7 and Hoechst. SEPT7 cages are indicated with arrowheads. S. flexneri afaE strain (hyperinvasive) was used to compare WT infection to S. flexneri ΔrfaC. Scale bar, 5 μm (inset, 2 μm). **d** Percentage of septin caged bacteria in HeLa cells infected for 3 h 40 min. Data represent the mean ± SEM from $n = 1695$ (afaE) and $n = 1565$ (ΔrfaC) S. flexneri cells distributed in 8 independent experiments. **\*\***$p = 0.0033$ by two-tailed Student's $t$-test.

surface is covered by septins (Supplementary Fig. 8a). In the case of S. flexneri ΔrfaL, septins cover the entire bacterial surface and are also dramatically enriched at one bacterial pole (i.e., similar to septin recruitment to S. flexneri WT) (Supplementary Fig. 8a).

Septin recruitment to all mutants with altered LPS is restored back to WT levels when expressed in trans from a plasmid [rfaL ($53.6 \pm 4.1\%$), galU ($56.9 \pm 9\%$), rfaC ($47.5 \pm 11.1\%$)] (Supplementary Fig. 8a, b).

**The kinetics of septin binding depends on the bacterial species and the amphipathic helix domain of SEPT6**. To explore differences between septin recruitment profiles of *S. flexneri* and *M. smegmatis*, we combined microfluidics (CellASICS ONIX2) with time-lapse fluorescence microscopy and observed real-time binding events. We monitored the rate of septin binding at the single-cell level from the first observed binding event until the mean msGFP–SEPT6 intensity on the bacterial surface plateaus. While *S. flexneri* WT mostly recruits septins to one bacterial pole, *M. smegmatis* is fully covered by septins in our in vitro system (Fig. 1c, f). Time-lapse experiments were therefore performed using *S. flexneri* Δ*rfaL* where septins fully cover the bacterial surface and are also dramatically enriched at one bacterial pole (Supplementary Fig. 8a). Strikingly, septins are recruited significantly faster to the surface of *M. smegmatis* than to the surface *S. flexneri* (Fig. 5a, b and Supplementary Movies 1 and 2). Using the time-dependent increase in msGFP–SEPT6 fluorescence signal as a reporter for septin binding events, we employed a kinetic model and extracted the binding rates ($k_b$) for *S. flexneri* Δ*rfaL* and *M. smegmatis*. From this, single-cell analysis shows that the rate of septin binding is significantly higher in *M. smegmatis* with $k_b = (20.0 \pm 0.8) \times 10^{-3} \, s^{-1}$ than in *S. flexneri* Δ*rfaL* with $k_b = (8.6 \pm 0.5) \times 10^{-3} \, s^{-1}$ (Fig. 5c, d). These data suggest that septins bind to the surface of *M. smegmatis* with a higher affinity than they bind to the surface of *S. flexneri*, likely due to differences in lipid composition. To ensure that septin binding to bacterial cells is physiological under our experimental conditions, we tested for septin cage recruitment to *E. coli* (MG1655). Consistent with our in vitro reconstitution quantifications (not performed in real time, Fig. 2c, d), these live-cell experiments confirmed that *E. coli* are not entrapped in septin cages (Supplementary Movie 3).

We employed the same single-cell approach to investigate the kinetics of septin binding of msGFP-SEPT6WT- and msGFP-SEPT6ΔAH-septin complexes to *S. flexneri* (Fig. 5c, d). In this case, single-cell analysis showed that msGFP-SEPT6WT-containing septin complexes bind *S. flexneri* Δ*rfaL* cells at a higher rate [$k_b = (8.6 \pm 0.5) \times 10^{-3} \, s^{-1}$] than msGFP-SEPT6ΔAH-containing complexes [$k_b = (6.8 \pm 0.3) \times 10^{-3} \, s^{-1}$)] (Fig. 5c, d and Supplementary Movies 1 and 4). These data show that the SEPT6 AH domain promotes the binding rate of septin complexes to the bacterial surface.

**IcsA disrupts lipopolysaccharide at the bacterial pole and promotes septin recognition of *S. flexneri***. To further investigate the role of IcsA in septin binding to the surface of *S. flexneri*, we tested for IcsA-septin (SEPT2–SEPT6–SEPT7) interaction using pull-down assays, but these experiments failed to show a direct interaction (data not shown). We then engineered double mutants lacking LPS components (i.e., Δ*rfaC*, Δ*galU*, Δ*rfaL*) and *icsA* (Δ*icsA*). In the case of *S. flexneri* Δ*rfaC*Δ*icsA*, we observed that septins can directly bind to the bacterial surface in the absence of IcsA (Fig. 6a, b). Consistent with this, bacterial sedimentation assays showed no differences in the amount of SEPT7 bound to the single Δ*rfaC* mutant as compared to *S. flexneri* Δ*rfaC*Δ*icsA* (Fig. 6c). In the case of *S. flexneri* Δ*galU*Δ*icsA*, we observed that septin recruitment is less homogeneous than the single Δ*galU* mutant, suggesting that the absence of IcsA can affect the distribution of septins on bacterial surfaces but not the overall recognition of bacterial cells by septins (Fig. 6a, b). In the case of *S. flexneri* Δ*rfaL*Δ*icsA*, the percentage of septin-recruiting bacteria was significantly reduced as compared to the single Δ*rfaL* mutant, suggesting that IcsA is important for bacterial recognition by septins when the outer core of LPS is present (Fig. 6a, b). Together, these data support a model in which (i) O-antigen chains and the outer core of LPS impose a physical barrier that masks the bacterial surface and prevents septin cage entrapment,

and (ii) IcsA disrupts LPS at the bacterial pole and creates pores to permit the interaction of septins with the bacterial surface.

To investigate how *S. flexneri* LPS can physically prevent septins from binding to the bacterial outer membrane, we advanced our in vitro reconstitution system using *S. flexneri* WT and Δ*rfaC*, and visualized septin filaments on the surface of bacteria by correlative cryo-light microscopy and cryoET. Remarkably, cryoET showed that septins are spaced 19.8 ± 0.7 nm from the membrane of *S. flexneri* WT (Fig. 6d, top panel, and Fig. 6e), yet in the absence of O-antigen as well as outer and inner core components of LPS, septin filaments are only spaced 8.9 ± 0.2 nm from the outer membrane of *S. flexneri* Δ*rfaC* (Fig. 6d, bottom panel, and Fig. 6e). To confirm that structures we observe by cryoET on the surface of bacteria are septin filaments, we imaged *S. flexneri* WT ($n = 9$ tomograms) and Δ*rfaC* ($n = 10$ tomograms) not incubated with septins in vitro; in this case, we did not observe any structures bound to the bacterial surface (Supplementary Fig. 9). These results demonstrate a role for LPS as a physical barrier preventing membrane-septin interactions and highlight a new mechanism in which LPS prevents bacterial recognition by cell-autonomous immunity.

## Discussion

How septins can recognize and assemble on bacterial cells for cage entrapment was not known. Here, we employed a cell-free reconstitution system to discover that septins, in the absence of additional host cell factors, recognize the bacterial surface of growing *S. flexneri* cells for cage entrapment. In agreement with septin cages being a crucial component of cell-autonomous immunity, we discover that *S. flexneri* LPS masks the bacterial surface and acts as a physical barrier that protects bacteria from septin cage entrapment. Our data suggest that IcsA disturbs LPS at the bacterial pole of *S. flexneri* and promotes septin recruitment to the bacterial surface. This model can explain why septin binding in vitro mainly occurs at one pole of the *S. flexneri* WT cell (i.e., where IcsA is located). According to this model, the SEPT6 AH domain would contribute by sensing curvature at the same bacterial pole where IcsA is located (and where septins are interacting with the bacterial surface). Moreover, septins possess other domains that may promote interactions with membranes, including their polybasic domain[22]. It is not currently known if septins can employ these different protein domains to bind bacterial surfaces, nor how this might influence septin-bacteria spacing.

In a next step, our in vitro reconstitution system can be used to screen for additional bacterial factors that promote or inhibit septin cage entrapment. Investigating the coordination of actin polymerization and septin assembly using bacterial surfaces can provide new insights into cytoskeletal crosstalk. We demonstrate that human SEPT6 encodes an AH domain that senses micron-scale membrane curvature and promotes *S. flexneri* septin cage entrapment; the precise role of the SEPT6 AH domain in antibacterial autophagy has not yet been tested. In the future it will be of great interest to implement SEPT9 into hetero-oligomers in vitro[40,41], and compare bacterial recognition by SEPT2–SEPT6–SEPT7 versus SEPT2–SEPT6–SEPT7-SEPT9 complexes. In addition, our in vitro reconstitution system using bacteria can be used to test the role of other septin features including their protein domains (e.g., GTP-binding, C-terminal coiled-coil, septin unique element -SUE-), post translational modifications (e.g., phosphorylation, ubiquitination, SUMOylation) and mutations associated with human disease[42]. The combination of our in vitro reconstitution system with cell-free extracts can inspire the discovery of unknown host factors modulating septin cage entrapment, and merging our purified septin complexes with autophagosomes reconstituted in vitro[43,44] may illuminate the precise role of septins in targeting bacteria to autophagy.

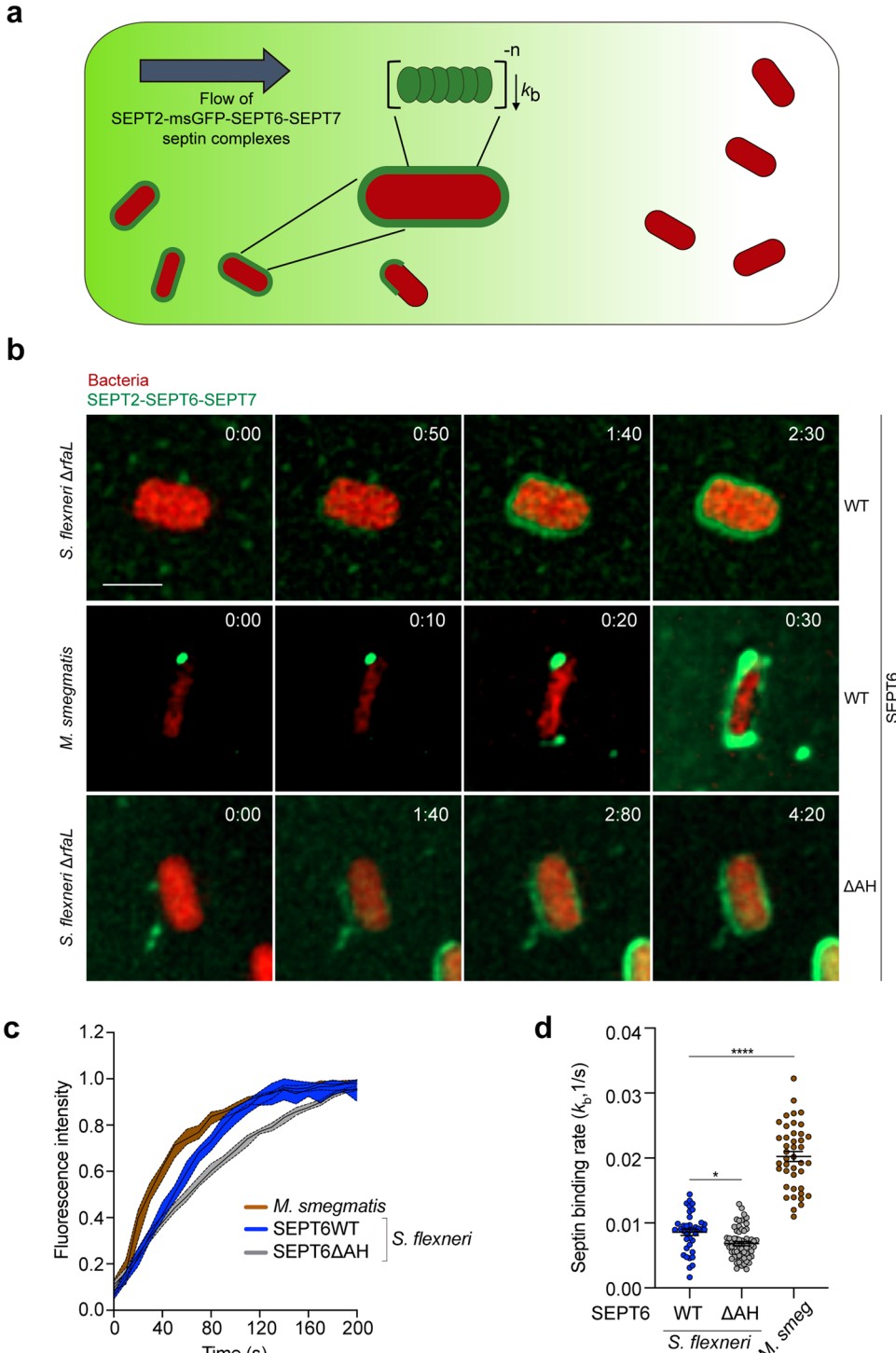

**Fig. 5 The kinetics of septin binding depends on bacterial species and the amphipathic helix domain of SEPT6. a** Scheme showing the protocol followed to calculate the binding rates of SEPT2–msGFP-SEPT6–SEPT7 septin complexes (green) to bacteria using microfluidics and fluorescence microscopy. **b** Representative time-lapse fluorescence microscopy showing the entrapment of *S. flexneri* Δ*rfaL* producing cytosolic mCherry (msGFP-SEPT6WT-complexes, top panel; msGFP-SEPT6ΔAH-complexes, bottom panel) and *M. smegmatis* (msGFP-SEPT6WT-complexes, middle panel) in septin cages (green) in vitro. Time is showed as min:s (top right corner). This experiment was performed 2 independent times. Scale bar, 2 μm. **c** Mean msGFP–SEPT6 intensity per cell over time showing binding of septin complexes to *S. flexneri* Δ*rfaL* mCherry (blue line, msGFP-SEPT6WT; gray line, msGFP-SEPT6ΔAH) and *M. smegmatis* DsRed (brown line). Mean intensity values are normalized by maximum intensity observed in each cell for visualization purposes. Data represent the median ± 95% confident interval of $n = 38$ (*S. flexneri* Δ*rfaL* + msGFP-SEPT6WT), $n = 60$ (*S. flexneri* Δ*rfaL* + msGFP-SEPT6ΔAH), and $n = 42$ (*M. smegmatis*) cells distributed in 2 independent experiments. **d** Extracted binding rates ($k_b$) of septin complexes to *S. flexneri* Δ*rfaL* mCherry and *M. smegmatis* DsRed in full septin cages in vitro. Data represent the mean ± SEM of $n = 38$ (*S. flexneri* Δ*rfaL* + msGFP-SEPT6WT), $n = 60$ (*S. flexneri* Δ*rfaL* + msGFP-SEPT6ΔAH), and $n = 42$ (*M. smegmatis* + msGFP-SEPT6WT) cells distributed in 2 independent experiments. \*$p = 0.0338$, \*\*\*\*$p < 0.0001$ by one-way ANOVA and Dunnett's post-test.

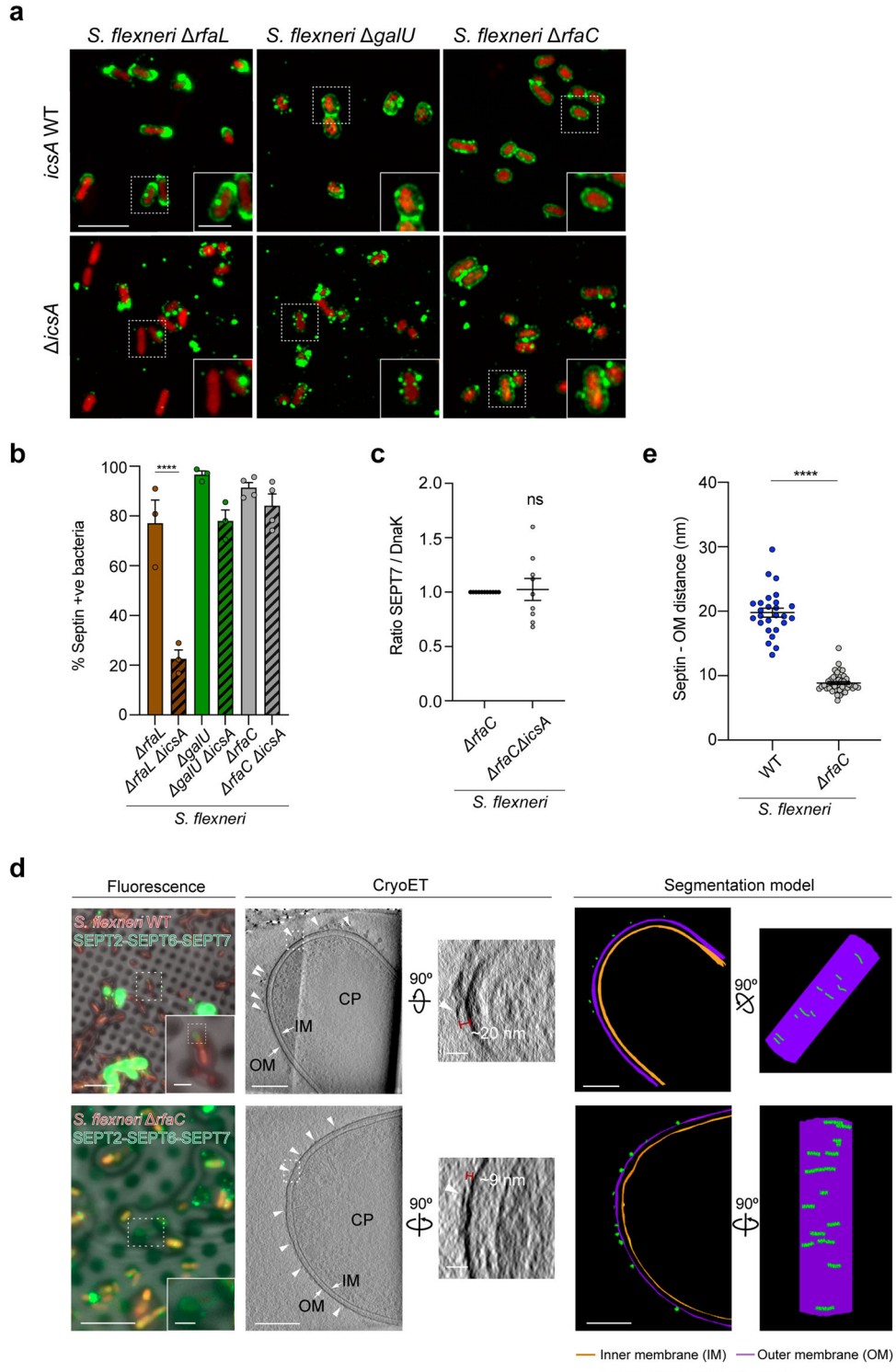

**Fig. 6 IcsA disrupts lipopolysaccharide and promotes septin recognition of *S. flexneri*. a** Airyscan confocal images comparing septin binding in the presence (top) or absence (bottom) of IcsA to *S. flexneri* Δ*rfaC*, Δ*rfaL*, and Δ*galU*. Scale bar, 5 μm. **b** Percentage of bacteria recruiting septins in vitro. Quantifications represent mean ± SEM from $n = 1022$ (Δ*rfaC*), $n = 1072$ (Δ*rfaC*Δ*icsA*), $n = 594$ (Δ*rfaL*), $n = 605$ (Δ*rfaL*Δ*icsA*), $n = 653$ (Δ*galU*) and $n = 761$ (Δ*galU*Δ*icsA*) *S. flexneri* cells distributed in 3 independent experiments. ****$p < 0.0001$ by one-way ANOVA and Tukey's post-test. **c** Bacterial sedimentation assays of samples from panel (**a**). Data represent mean ± SEM from 4 independent blots. ns, $p = 0.665$ by two-tailed Mann–Whitney's test. **d** Representative correlative cryo-light microscopy and cryo-ET images of *S. flexneri* WT (top, accession No. #EMD-12571) or Δ*rfaC* (bottom, accession No. #EMD-12579) showing bound septins in vitro (arrowheads and green segmentation). Images shown correspond to a slice of 10.8-nm thickness (*S. flexneri* WT) and a slice of 11.0-nm thickness (*S. flexneri* Δ*rfaC*). CP, cytoplasm; OM, outer membrane; IM, inner membrane. This experiment was performed 2 independent times. Scale bars, fluorescence microscopy, 10 μm (inset, 2 μm), cryoET, 200 nm (inset, 50 nm). **e** Measured distance between septin filaments and the *S. flexneri* outer membrane. Data correspond to mean ± SEM from $n = 26$ (*S. flexneri* WT) and $n = 46$ (*S. flexneri* Δ*rfaC*) filaments from 3 independent tomograms. ****$p < 0.0001$ by two-tailed Student's *t*-test.

Combining our in vitro reconstitution system with cryoET has revealed, for the first time, that septins assemble as filaments on the surface of bacterial cells. Unfortunately, it is highly challenging to visualize septin filaments across the entire bacterial surface due to limited tilting range (or "missing wedge") and projection imaging in cryoET. Future work using our in vitro reconstitution system with Focused Ion Beam milling, cryoET and sub-tomogram averaging can significantly advance our understanding of how septins interact with bacterial surfaces at the molecular level.

Finally, our data using *S. flexneri* and mycobacteria suggest that septins can distinguish between different bacterial species, likely because the lipid composition of the *S. flexneri* bacterial surface differs from that of mycobacteria. In addition, *S. flexneri* possesses a layer of LPS that can prevent septin interactions with the bacterial surface. The ability of septins to differentiate between bacterial cells (e.g., growing or not) and closely related species (e.g., *S. flexneri* versus *E. coli*) may have great translational impact for human health, for example as a novel approach to identify and entrap pathogenic bacteria in vitro. Unlike other reports using supported lipid bilayers[21,45,46], it is exciting to consider the use of bacteria to investigate septin biology and how it will deliver fundamental understanding of bacterial surfaces.

## Methods

**Reagents**. The following antibodies were used: rabbit anti-SEPT7 (1:1000 dilution, #18991, IBL), mouse anti-DnaK (1:5000, #ADI-SPA-880, Enzo), mouse anti-GFP (1:4000, #ab1218, abcam), rabbit anti-GBP1 (1:1000 #15303-1-AP, Proteintech), mouse anti-GAPDH (1:1000 or 1:2000, #ab8245, Abcam), goat HRP-conjugated anti-mouse (1:5000 or 1:10,000, #P0260, Dako), goat HRP-conjugated anti-rabbit (1:5000 or 1:10,000, #P0448, Dako), Alexa-555-conjugated anti-rabbit antibody (1:500, #10082602, ThermoFisher Scientific), Alexa-647-conjugated anti-rabbit antibody (1:500, #A27040, ThermoFisher Scientific). The following dyes and drug were used: Hoechst (1:300, #H3570, ThermoFisher Scientific), Alexa-488-conjugated phalloidin (1:300, #A12379, ThermoFisher Scientific), IFNγ (#285-IF, R&D Systems).

**Bacterial strains and culture conditions**. The bacterial strains and plasmids described in this study are listed in Supplementary Table 1. Unless indicated *Shigella flexneri* 5a str. M90T was used as control throughout the manuscript. *Shigella* strains were grown in trypticase soy broth (TCS)-agar containing 0.01% (w/v) congo red to select for red colonies, indicative of a functional T3SS. Conical polypropylene tubes (#CLS430828, Corning) containing 5 ml of TCS were inoculated with individual red colonies of *S. flexneri* and were grown ~16 h at 37 °C with shaking at 200 rpm. The following day, bacterial cultures were diluted in fresh prewarmed TCS (1:50 or 1:100 v/v), and cultured until an optical density (OD, measured at 600 nm) of 0.6. To grow *S. flexneri* Δ*rfaL* (#QCC33417.1), Δ*galU* (#QCC31905.1), Δ*rfaC* (*waaC*, #QCC33416.1), Δ*rfaL*Δ*icsA*, Δ*galU*Δ*icsA*, Δ*rfaC*-Δ*icsA*, Δ*icsB*, and Δ*mxiD* TCS was supplemented with 50 μg/ml of kanamycin. To maintain the plasmids encoding *icsA*, IcsA$^{507-620}$-mCherry, *mCherry* or *afaE*, TCS was supplemented with 100 μg/ml of carbenicillin.

IcsA$^{507-620}$-mCherry, is a cytosolic derivate of IcsA that contains the targeting sequence for polar localization (gift from Marcia Goldberg, Harvard Medical School, Boston, MA). The production of this protein was induced with 1 mM Isopropyl β- d-1-thiogalactopyranoside (IPTG) 15 min prior to image analysis.

*Escherichia coli* strains were grown in Lysogeny-Broth (LB) in conical polypropylene tubes at 37 °C with shaking at 220 rpm. *E. coli* DH5α was used to purify pKD46 and pKD4 plasmids, and LB was supplemented with 100 μg/ml of carbenicillin or 50 μg/ml of kanamycin, respectively. *E. coli* BL21-DE3- (ThermoScientific) was used for protein production and purification, and for the in vitro assays. We selected this strain of *E. coli* because it lacks the outer membrane protease OmpT, known to cleave IcsA[47].

*Mycobacterium smegmatis* str. mc$^2$155 (gift from Gerald Larrouy-Maumus) was grown in TCS in conical polypropylene tubes at 37 °C with shaking at 220 rpm. *Mycobacterium marinum* str. M[16] was grown in 7H9 medium (supplemented with ADC Enrichment medium, 0.2% glycerol, hygromycin 50 μg/ml) in 75 cm vented flasks (#10170961, ThermoFisher Scientific) at 28 °C in static conditions.

All bacterial stocks were stored in 10% glycerol at −80 °C.

**Design of bacterial mutants**. Primers used in this study were designed using Benchling (https://benchling.com) or NEBuilder Assembly Tool (https://nebuilder.neb.com/#!/) and are listed in Supplementary Table 2. *S. flexneri* mutants were engineered using λ-Red-mediated recombination[48]. In brief, kanamycin resistance-encoding DNA cassettes were amplified using pKD4 plasmid as template and primers containing 50 bp nucleotides homologous to the site of insertion. Resulting DNA fragments were electroporated in *S. flexneri* electrocompetent cells producing λ-Red recombinase and plated in TSA plates supplemented with 0.01% of congo red and 50 μg/ml of kanamycin. To engineer double mutants, the kanamycin resistance-encoding DNA cassettes substituting deleted Δ*rfaL*, *galU*, or *rfaC* genes were removed by transforming *S. flexneri* Δ*rfaL*, Δ*galU*, and Δ*rfaC* mutant strains with pCP20 plasmid (that encodes the yeast *flp* recombinase)[48]. *icsA* gene was subsequently deleted as mentioned. All strains were verified by PCR.

**Complementation of bacterial mutants**. To complement *S. flexneri* Δ*rfaL*, Δ*galU*, and Δ*rfaC* mutant strains, *rfaL*, *galU*, and *rfaC* genes were cloned in a pFUS-PBAD plasmid (encoding an arabinose inducible promoter and an optimized Shine–Dalgarno assembly)[49] by Gibson assembly. Briefly, *rfaL*, *galU*, and *rfaC* genes were PCR-amplified from *S. flexneri* M90T genome using the primers *rfaC*-fw and *rfaC*-rv, *rfaL*-fw and *rfaL*-rv, and *galU*-fw and *galU*-rv, respectively. pFUS-PBAD backbone was PCR-amplified using the primers *pFUS-Gib-fw* and *pFUS-Gib-rv*. Gibson assembly was performed at 50 °C for 30 min using the HiFi DNA Assembly Master Mix (#E2621L, New England Biolabs). Resulting plasmids prfaL, pgalU, and prfaC were transformed into *S. flexneri* Δ*rfaL*, Δ*galU*, and Δ*rfaC*, respectively. Before introducing complementing plasmids, the kanamycin resistance-encoding gene (substituting Δ*rfaL*, Δ*galU*, and Δ*rfaC*) was removed using pCP20 plasmid as described above.

**SDS/EDTA sensitivity assays**. Conical polypropylene tubes containing 5 ml of TCS were inoculated with individual colonies and grown for ~16 h at 37 °C with shaking at 200 rpm. The following day, bacterial cultures were diluted in fresh prewarmed TCS (1: 100 v/v) and cultured until an OD$_{600}$ of 0.6 at 37 °C with shaking at 200 rpm. Bacteria were washed 2× PBS pH 7.4, serially diluted and 5 μl spots were plated on LB-agar or LB-agar containing 0.05% SDS and 0.28 mM EDTA plates. EDTA chelates divalent ions important for LPS stability. SDS is a cationic detergent that inserts into the bacterial outer membrane.

**Mammalian cell culture**. HeLa (ATCC CCL-2) cells were grown at 37 °C and 5% CO$_2$ in Dulbecco's modified Eagle medium (DMEM, GIBCO) supplemented with 10% fetal bovine serum (FBS, Sigma-Aldrich). GFP–SEPT6[50], msGFP–SEPT6, and msGFP–SEPT6ΔAH-producing HeLa cells were grown as mentioned above in DMEM supplemented with 10% FBS and 2 μg/ml of puromycin.

The production of GBP1 was tested by treating HeLa cells with different concentrations of IFNγ for 24 h (Supplementary Fig. 4a). Based on our results we advanced with 100 ng/ml IFNγ for 24 h.

**Infection of human cells**. In total, $9 \times 10^4$ HeLa cells were seeded in 6-well plates (Thermo Scientific) containing 22 × 22 mm glass coverslips 2 days before the infection. Bacterial cultures were grown as described and cell cultures were infected with *S. flexneri* strains at a multiplicity of infection (MOI, bacteria:cell) of 5:1 (*afaI*, Δ*rfaC*) or 100:1 (srv. 5a str. M90T). In the case of *S. flexneri* M90T, bacteria and cells were immediately centrifuged at 110 × *g* for 10 min at room temperature (RT). Then, plates were placed at 37 °C and 5% CO$_2$ for 30 min. Infected cultures were washed 2× with phosphate-buffered saline (PBS) pH 7.4 and incubated with fresh DMEM containing 10% FBS and 50 mg/mL gentamicin at 37 °C and 5% CO$_2$ for 3 or 4 h.

**Design of recombinant septins**. Recombinant human septins SEPT2–msGFP-SEPT6–SEPT7 and SEPT2–msGFP–SEPT6ΔAH–SEPT7 complexes were purified as previously described in ref. [51].

Plasmids encoding *SEPT2* (pnEA-vH-SEPT2), and *SEPT6* and *SEPT7* (pnCS-SEPT6–SEPT7) were used[14]. For in vitro reconstitution assays we engineered a fusion protein formed by a N-terminal monomeric superfolder version of green fluorescent protein (msGFP) and human SEPT6. We selected msGFP because it has been previously employed to labeled bacterial FtsZ[14], a filament forming protein, without affecting its function. The msGFP gene was PCR-amplified from a pCU19 plasmid using a primer (*Spe-SD-GFP-5*) encoding the original Shine–Dalgarno sequence of pnCS-SEPT6–SEPT7 and a SpeI restriction site, and a second primer (*TEV-Xba-GFP-3*) encoding a linker sequence (GGSRENLYFQGSG) and a XbaI site. Then, the N-terminus of *SEPT6* was amplified from pnCS-SEPT6–SEPT7 using a primer (*Xba-TEV-S6-5*) encoding the linker sequence and a XbaI site and a second primer encoding a ClaI site (*Pst-Cla-S6*). DNA fragments from both PCRs were mixed and PCR-amplified using primers *Spe-SD-GFP-5* and *Pst-Cla-S6*, digested with SpeI and ClaI enzymes and ligated with a SpeI/ClaI digested pnCS-SEPT6–SEPT7 plasmid, generating the pnCS-msGFP–SEPT6–SEPT7 plasmid.

To purify msGFP–SEPT6ΔAH we generated the pnCS-msGFP–SEPT6ΔAH–SEPT7 plasmid by Gibson assembly. The vector backbone was PCR-amplified from pnCS-msGFP–SEPT6–SEPT7 using the primers *pnCS-fwd* and *pnCS-rv*. The insert containing *msGFP–SEPT6ΔAH* was PCR-amplified from pLVX-msGFP–SEPT6ΔAH using the primers *msGFP-S6ΔAH-fwd* and *msGFP-S6ΔAH-fwd* (see paragraph below). Gibson assembly was performed at 50 °C for 30 min using the HiFi DNA Assembly Master Mix.

To engineer HeLa cells stably producing msGFP–SEPT6 or msGFP–SEPT6ΔAH (lacking the AH domain) we designed the plasmids pLVX-msGFP–SEPT6 and

pLVX-msGFP–SEPT6ΔAH by Gibson Assembly. First, a DNA fragment encoding msGFP–SEPT6 was PCR-amplified using oligos *msGFP–SEPT6-fwd* and *msGFP–SEPT6-rv* and the pnCS-msGFP–SEPT6–SEPT7 plasmid as template (described above). In the case of msGFP–SEPT6ΔAH the construct was amplified in two steps: (1) a DNA fragment containing *msGFP* and the N-terminus (just before the AH-encoding sequence) of *SEPT6* was PCR-amplified using the primers *msGFP-N-SEPT6-fwd* and *msGFP-N-SEPT6-rv* using pnCS-msGFP–SEPT6–SEPT7 as template; (2) a DNA fragment containing C-terminus of SEPT6 was PCR-amplified using the primers *C-SEPT6-fwd* and *C-SEPT6-rv* from pnCS-msGFP–SEPT6–SEPT7. Then, pLVX vector was PCR-amplified using the primers *pLVX-fwd* and *pLVX-rv*, and the DNA fragment containing *msGFP–SEPT6ΔAH* generated in the previous step was used as template for PCR amplification using the oligos *msGFP–SEPT6-fwd* and *msGFP–SEPT6-rv*. Gibson assembly was performed at 50 °C for 30 min.

**Development of transgenic cell lines.** HeLa cells were infected with the lentiviral expression vector pLVX-msGFP–SEPT6 or pLVX-msGFP–SEPT6ΔAH to generate cells stably expressing each construct as previously described[15]. Briefly, $1.5–2 \times 10^6$ HEK293FT cells were seeded on 6-well plates and co-transfected with 1.6 μg of pLVX-msGFP–SEPT6 or pLVX-msGFP–SEPT6ΔAH, 1.2 μg of psPAX2, and 0.5 μg of pMD2.g vectors for 6 h using Lipofectamine 2000 (ThermoScientific). Then, the medium was changed with fresh prewarmed DMEM supplemented with 10% FBS. Supernatants from transfected HEK293FT cells containing the lentiviral particles were collected 24 and 48 h after transfection. Different volumes of the lentiviral solution (50–500 μl) were added to $10^5$ HeLa cells seeded on 6-well plates the day before and were incubated for 3 days. Stably producing msGFP–SEPT6 or msGFP–SEPT6ΔAH cells were selected by adding 1 μg/ml of puromycin to the culturing media.

**In vitro reconstitution of septin cages.** *S. flexneri* cultures were grown 16 h in conical polypropylene tubes containing 5 ml of M9-Tris (50 mM Tris-HCl pH 8, 50 mM KCl, 0.5 mM $MgCl_2$, 10 mM $CaCl_2$, 100 mM $MgSO_4$) salts supplemented with a mix of nutrients (45 μg/ml L-methionine, 20 μg/ml L-tryptophan, 12.5 μg/ml nicotinic acid, 10 μg/ml vitamin B1, 1% glucose, 0.5% casein hydrolysate, 0.1% fatty acid-free BSA) -M9-Tris-CAA- at 37 °C with shaking at 200 rpm. This optimized reaction solution was based on an optimized minimal growth medium for *Shigella*[52,53] and a buffer that permits septin assembly into filaments/bundles[21]. The following day, bacterial cultures were diluted in 10 ml of fresh prewarmed M9-Tris-CAA (1:100 v/v) in conical polypropylene tubes and cultured until an $OD_{600}$ of 0.6. 1.2 ml of bacterial cultures were centrifuged in Low Protein Binding tubes (ThermoFisher Scientific) at $800 \times g$ for 2 min at RT and the supernatant was removed. To measure binding of septins to *S. flexneri*, bacterial pellets were resuspended in 100 μl of in vitro reconstitution solution [M9-Tris-CAA supplemented with 240 nM of septin complex SEPT7/msGFP–SEPT6/SEPT2 and 1 mM dithiothreitol (DTT)]. We used 240 nM of septin complexes throughout the manuscript, except for live-cell experiments involving microfluidics, where we used 2.4 μM of septin complexes. Purified septins in septin storage buffer (50 mM Tris pH 8, 300 mM KCl, 5 mM $MgCl_2$, and 5 mM DTT) were thawed on ice, diluted, and added to the in vitro reconstitution solution yielding a final buffer composition of 50 mM Tris pH 8, 50 mM KCl, 0.5 mM $MgCl_2$, and 1 mM DTT. Low Protein Binding tubes containing the bacteria in the in vitro reconstitution solution were placed in opaque conical polypropylene tubes and incubated at 37 °C with shaking at 220 rpm for 2 h until equilibrium was reached. Following the in vitro reconstitution reaction, samples were immediately placed on ice. To remove unbound septins, samples were centrifuged at $800 \times g$ at 4 °C for 1.5 min. Supernatant was carefully removed, bacterial pellet containing bound septins resuspended in 300 μl of ice-chilled M9-Tris-CAA buffer and centrifuged at $800 \times g$ at 4 °C for 2 min. This step was repeated one more time, to ensure removal of unbound septins, and pellets were finally resuspended in 100 μl of ice-chilled M9-Tris-CAA buffer. These samples were then used for bacterial sedimentation assays, confocal imaging, or cryoET.

For other conditions, the protocol had the following modifications: (1) Bacteria were grown in TCS (*S. flexneri, M. smegmatis*), LB (*E. coli*), or 7H9 (*M. marinum*) instead of M9-Tris-CAA as mentioned before, and washed twice in septin buffer (50 mM Tris pH 8, 50 mM KCl, 0.5 mM $MgCl_2$) to remove traces of growth medium; (2) the in vitro reconstitution solution was 50 mM Tris pH 8, 50 mM KCl, 0.5 mM $MgCl_2$, 1 mM DTT, and 0.1% fatty acid-free BSA.

To complement Δ*rfaL*, Δ*galU*, and Δ*rfaC* in the in vitro reconstitution system, *S. flexneri* cultures were grown 16 h in conical polypropylene tubes containing 5 ml of M9-Tris-CAA supplemented with 0.05% L-arabinose for 16 h at 37 °C and 200 rpm. The following day, bacterial cultures were diluted in 10 ml of fresh prewarmed M9-Tris-CAA (1:100 v/v) containing 0.3% L-arabinose in conical polypropylene tubes and cultured until an $OD_{600}$ of 0.6. Bacterial cultures were processed as mentioned above.

**Bacterial sedimentation assays.** In vitro reconstituted samples were prepared as mentioned above and mixed with Laemmli buffer[54]. Proteins were resolved by SDS-PAGE and blotted against SEPT7 and DnaK (used as loading control). Of note, we blotted the same membranes against both antibodies. To correlate

confocal imaging with bacterial sedimentation assays results, we employed the same samples for both types of experiments. Densitometry of the bands was performed in Fiji. The amount of septins bound to bacteria was quantified as the ratio SEPT7/DnaK and normalized to the control sample (*S. flexneri* M90T incubated with SEPT2–msGFP-SEPT6–SEPT7).

**Western blotting.** Samples were lysed in Laemmli buffer and incubated at 95 °C for 10 min. Proteins were resolved in 10 or 12% SDS–polyacrylamide gels and transferred to polyvinylidene difluoride membranes (PVDF, #IPVH00010, MerckMillipore). PVDF membranes were incubated with the primary antibody for 1 h 30 min at RT. Primary antibodies were diluted (1 μg/ml of anti-SEPT7, 0.2 μg/ml of anti-DnaK, or 1 μg/ml of anti-GFP) in blocking solution (75 mM Tris-HCl pH 8.8, 150 mM NaCl, 0.1% Tween20) supplemented with 3% fatty acid-free milk. PVDF membranes were washed 3× 5–7 min in blocking solution at RT and incubated with secondary goat HRP-conjugated antibodies for 1 h at RT. PVDF membranes were washed 3× 5–7 min in blocking solution at RT and developed using Pierce™ ECL plus western blotting substrate.

**Immunostaining and fluorescence microscopy.** Infected or uninfected cells were washed 3× with PBS pH 7.4 and fixed 15 min in 4% paraformaldehyde (in PBS) at RT. Fixed cells were washed 3× with PBS pH 7.4 and subsequently permeabilized 5 min with 0.1% Triton X-100 (in PBS). Cells were then washed 3–6× in PBS and incubated 1 h 30 min with primary anti-SEPT7 antibody diluted in PBS supplemented with 0.1% Triton X-100 and 1% bovine serum albumin. Cells were then washed 3–6× in PBS and incubated 45 min with Alexa-555-conjugated anti-rabbit secondary antibody diluted 0.1% Triton X-100 (in PBS). Cells were then washed 3–6× in PBS and incubated 40 min with a solution of 0.1% Triton X-100 (in PBS) containing Hoechst and Alexa-488-conjugated phalloidin where indicated. Coverslips were placed on glass slides and samples were preserved with aqua polymount mounting medium (ID#18606, Polyscience).

Fluorescence microscopy was performed using a 63×/1.4 C-Plan Apo oil immersion lens on a Zeiss LSM 880 confocal microscope driven by ZEN Black software (v2.3). Microscopy images were obtained using z-stack image series taking 8–16 slices.

Confocal images were processed using Airyscan processing (Weiner filter) using "Auto Filter" and "3D Processing" options.

**Correlative cryo-light microscopy and cryo-electron tomography.** In vitro reconstitution samples were mixed with 10 nm BSA-coated colloid gold particles at a ratio 1:5 (bacteria:gold particles) and placed on 200 mesh Quantifoil Copper grids R 2/2 (*M. smegmatis*) or Quantifoild Gold Finder grids R 2/2 (*S. flexneri*). Grids were vitrified using a Vitrobot Mark IV (Thermo Fisher). Only grids with appropriate ice thickness and good bacterial distribution were used for subsequent tomogram data collection.

For correlative cryo-light and cryo-electron microscopy, vitrified grids containing *S. flexneri* WT or Δ*rfaC* were transferred to a cryo-stage (CMS-196, Linkam Scientific) and imaged using a 100× EC Epiplan-NeoFluar objetive on a Zeiss AxioImager Z2 driven by ZEN Blue software. Fluorescence images of selected areas were manually correlated with the corresponding TEM square montages using SerialEM (Mastronarde, 2005) when setting up cryo-ET data collection.

CryoET datasets of septin caged *S. flexneri* WT or *M. smegmatis* were collected using a Titan Krios (Thermo Fisher) electron microscope operating at 300 kV equipped with an energy filter and a K3 Summit camera (Gatan Inc.), at a nominal magnification of 33,000 (an effective pixel size of 2.68 Å). CryoET datasets of septin-free *M. smegmatis* or septin caged *S. flexneri* Δ*rfaC* were collected using a Titan Krios (Thermo Fisher) electron microscope operating at 300 kV equipped with an energy filter and a K2 Summit camera (Gatan Inc.), at a nominal magnification of 53,000 (an effective pixel size of 2.75 Å). All tilt series were collected using SerialEM with a defocus value of −8 μm. Tilt series collection was performed using a bidirectional tilt scheme from −10° to +60° and then −12° to −60° in 2° incremental steps. The dose rate of each tilt was ~2.1 e⁻/Å² and the total accumulated dose was ~130 e⁻/Å².

Collected tilt series were aligned, reconstructed into 3D volumes using IMOD[55]. Segmentations were performed in IMOD using the final tomograms at a binning factor of $4 \times 4$. The contrast of selected tomograms was further enhanced using the deconvolution filter *tom_deconv*[56].

Distances between septin filaments and bacterial surfaces were measured on the segmentation models using a custom python script (Supplementary Software). Briefly, segmented models of septin filaments and bacterial surfaces were separately converted into coordinate files using *model2point* in IMOD. Then, distances between each point of septin filament model and all points of the bacterial surface model were calculated. The minimal distance of each measurement was used for statistics analysis.

All tomograms presented in this manuscript were deposited in The Electron Microscopy Data Bank (EMDB). The accession number for each tomogram is specified in the corresponding figure legend.

**Time-lapse fluorescence microscopy using microfluidic devices**. To follow the recruitment of septins to *S. flexneri* and *M. smegmatis* we combined time-lapse fluorescence microscopy with the Cell ASICS ONIX2 (MERCK-Millipore) microfluidic device. *S. flexneri* Δ*rfaL* or *M. smegmatis* were grown until $OD_{600} = 0.6$ in M9-Tris-CAA buffer or TCS, respectively. To prevent the adsorption of septin to the microfluidic channels, the Cell ASICS microfluidic chambers were passivated before loading bacteria with 5% TritonX-100 for 40 min at a flow rate of 1 psi (0.069 bar). TritonX-100 was loaded on wells 6 and 8. We then washed the chamber by two consecutive washing steps using $H_2O$ milliQ water and M9-Tris-CAA loaded on wells 6 and 8. Both washing steps were run for 40 min at a flow rate of 1 psi. We loaded well 8 with 50 µl of $20 \times 10^6$ bacterial cells/ml and flowed them into the microfluidic chamber using the default Cell Loading program. Before loading *M. smegmatis*, bacterial cells were washed once with M9-Tris-CAA. Cell density was confirmed by microscopy. We removed bacterial cells from well 8, washed it once with M9-Tris-CAA, and loaded it with 120 µl of 2.4 µM SEPT2–msGFP-SEPT6WT–SEPT7 or SEPT2–msGFP–SEPT6ΔAH–SEPT7 septin complexes (suspended in M9-Tris-CAA buffer). Wells 1–6 were loaded with 120 µl of M9-Tris-CAA buffer. M9-Tris-CAA was flowed from well 1 for 2 h at 37 °C to let bacteria grow and cells were imaged every 5 min using an AxioObserver Z1 fluorescence microscope driven by ZEN Blue v2.3 software (Carl Zeiss). Septins were then flowed from well 8 at 1 psi and bacterial cells were imaged every 10 s (using 10 z-stacks) to visualize septin recruitment over time.

**Image analysis of time-lapse movies and binding rate calculation**. To detect bacterial cells, we used cytoplasmic mCherry and DsRed signal for *S. flexneri* and *M. smegmatis*, respectively. Images were filtered using a 2D Gaussian filter (with a standard deviation of 0.5 pixels) to remove pixel noise. Cell boundaries were detected from maximum projections of z-stacks using the MATLAB-based cell segmentation tool Morphometrics[57]. Acquired cell contours were corrected to be equally spaced (0.5 pixels, 52.5 nm). Next, corrected sample points ($X_i$) were used to extract msGFP–SEPT6 signal on the bacterial surface.

First, msGFP–SEPT6 images were denoised with a Gaussian filter (as above) and a z-stack projection of maxima was applied. To avoid missing msGFP–SEPT6 signal (due to small displacements between cell contour and the GFP signal), fluorescent intensities were interpolated at boundary points $X_i$ and four other sets of points that lie perpendicular to the boundary and spaced 0.5 pixel, covering 1 pixel inward and outward of the detected contour[58]. We then computed the local background value for each cell at each time point to account for differences in the distribution of msGFP–SEPT6 signal in the field of view. Local background is defined as the median GFP pixel intensity value of the subregion of $7 \times 7$ microns with an origin at the cell's center. We acquired mean GFP intensity per cell ($I_k(t)$, where $k$ denotes cell number and $t$ time point) by subtracting local background from the mean of interpolated intensities. Due to the time lag between arrival of septin molecules in the field of view and the first observable recruitment event, we defined a $t_0$ for each cell where the $I_k(t_0)$ value is higher than local background fluorescence intensity. The rate of binding is calculated by considering data points appearing from $t_0$ (also called 'first observable recruitment event') until $t_f$, where the mean intensity plateaus.

We used $I_k(t)$ data points to fit a simple kinetic model assuming exponential kinetics. We extracted binding rates for each cell by fitting an exponential curve in the form $b(t) = a_1 - a_2 e^{-k_b \cdot t}$.

We used custom-written MATLAB (v2019a) scripts for image analysis and rate calculation (Supplementary Software).

**Quantification and statistical analysis**. Image processing and quantifications were performed in Fiji.

Where possible fluorescence microscopy images were randomized using the plugin for Fiji Filename_Randomizer.

Statistical analysis was performed in GraphPad Prism (v8.4, La Jolla, USA). Data represent the mean ± standard error of the mean (SEM). Fold changes were calculated from each independent experiment and the mean ± SEM are given in the text. When data were normalized to the control sample, the mean of the control was calculated from all independent experiments and the normalized values were calculated as the ratio (sample value)/(control mean value). A Student's *t*-test (two-tailed) or one-way ANOVA was used to test for statistical significance, with $p < 0.05$ considered as significant. All statistical details including statistical tests, significance, value of the number of experimental replicates, and bacterial cells quantified can be found in the figure legends.

All figures were designed using Adobe Illustrator CC 2018.

**Reporting summary**. Further information on research design is available in the Nature Research Reporting Summary linked to this article.

## Data availability

All data are included in the manuscript. Source data are provided with this paper. Materials can be obtained from the corresponding authors upon request. The following tomograms were deposited in EMDB: *M. smegmatis* + septins EMD-12562 (Fig. 1g), *M. smegmatis* NO septins EMD-12565 (Supplementary Fig. 2c), *S. flexneri* WT+ septins EMD-12571 (Fig. 6d), *S. flexneri* Δ*rfaC* + septins EMD-12579 (Fig. 6d), *S. flexneri* WT

NO septins EMD-12578 (Supplementary Fig. 9), and *S. flexneri* Δ*rfaC* NO septins EMD-12580 (Supplementary Fig. 9). Source data are provided with this paper.

## Code availability

Custom phyton (Supplementary Software related to Figs. 1h and 6e) and MATLAB (Supplementary Software related to Fig. 5) scripts were deposited in Github (https://github.com/xujweth/septin_scripts).

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

## Acknowledgements
We thank Mostowy lab members, Harry Low and Morgan Beeby for helpful discussions. We thank Kevin Cannon for his help to identify the SEPT6 amphipathic helix. We thank Gerald Larrouy-Maumus for providing *M. smegmatis*. We thank Haig Alexander Eskandarian for providing *M. smegmatis* DsRed. We thank ScopeM for cryo-EM instrument access at ETH Zürich. We thank Jannik Hugener for his help with correlative cryo-light microscopy and cryoET. D.L.-M. was funded by the European Union's Horizon 2020 research and innovation program under the Marie Skłodowska-Curie grant agreement No. H2020-MSCA-IF-2016-752022. Work in the M.P. Laboratory is supported by the Swiss National Science Foundation (No. 31003A_179255), the European Research Council (No. 679209), and the NOMIS Foundation. Work in the S.M. Laboratory is supported by a European Research Council Consolidator Grant (No. 772853-ENTRAPMENT), Wellcome Trust Senior Research Fellowship (206444/Z/17/Z), and the Lister Institute of Preventive Medicine.

## Author contributions
D.L.-M. and S.M. conceived the study and wrote the manuscript; all other authors commented on the manuscript. D.L.-M., J.X., G.Ö.G., M.P. and S.M. interpreted the data. D.L.-M. performed most experiments. J.X. performed cryo-ET/cryoLM experiments and D.L.-M. helped with data processing. G.Ö.G. analyzed microfluidics data and helped with AH experiments. A.O. helped design and test stable cell lines.

## Competing interests
The authors declare no competing interests.
