## [Peer Review File · Nature Communications]

REVIEWER COMMENTS

Reviewer #1 (Remarks to the Author):

Summary: In this study, Lobato-Marquez et. al. continue their excellent work on the role of septins in innate immunity by shedding light onto the mechanisms of bacterial cage formation by the septin cytoskeleton. They developed a powerful in vitro reconstitution platform and use this tool to show bacterial factors required for septin binding and cage formation. They have also identified an amphipathic helix within the C-terminus of Sept6, which is required for septin-mediated entrapment of *S. flexneri* in infected HeLa cells, providing further mechanistic insight into how septins interact with membranes during infection. Overall, I find this manuscript to be of high interest to the field.

I had no concerns with how the majority of experiments were designed / performed, however there were several points in the manuscript where I find it important that the authors elaborate on their data. Therefore, most of my comments can be addressed by revising the text.

"Major" comments:

1.) Regarding lines 98-102 " Both mycobacterial species are highly recognized by septins in vitro (67.8 ± 8.1 % *M. smegmatis* and 65.2 ± 8.8 % *M. marinum*) and in most cases septins assemble into cage-like structures that cover the entire surface of mycobacterial cells (Fig. 1f). This situation contrasted with *S. flexneri*, where septin binding is dependent on bacterial growth and mainly occurs at one pole of the bacterial cell."

This is very interesting. Can the authors speculate as to why this might be the case? Might it have to do something with the respective affinity for septins and a given bacteria? It might not even be a different mechanism (as the authors propose in the discussion), as one might speculate that it takes a shorter amount of time for septins to bind and mature to fully entrap *M. smegmatis* and *M. marinum* than *S. flexneri*. It might be fruitful to perform some time-lapse imaging to watch initial septin-bacteria binding and evolution into cage formation between these organisms to make this claim. Moreover, it might be worth measuring the percentage of bound septins between these organisms?

2.) Lines 108-110: "... septin filaments are spaced 15.9 nm from the mycobacterial outer membrane, a distance that may present the mycomembrane rich in mycobacterial lipids"

This reader is unsure exactly what this means. Does this suggest that septins not in direct contact with the membrane? Is the importance of the cardiolipin to recruit some other cellular factor to the cell surface in order to bind septins? Could it potentially be IcsA? It would be helpful if the authors could clarify and further expand on this point.

3.) The authors find that by reconstituting septins lacking the Sept6 AH domain and *S. flexneri*, there was no significant difference in binding relative to wild type. However, when the AH truncation is introduced into HeLa cells, the authors observe a significant reduction in the number of septin cages entrapping *S. flexneri*. This data is very interesting and is fairly consistent with what was observed in the previous study that identified an AH within the budding yeast septin Cdc12 (Cannon et. al., 2019). In this study, septins lacking the AH domain bound to all membrane curvatures when reconstituted in vitro, however in vivo, septins lacking the AH domain failed to localize to sites of curvature. Maybe it's unsurprising that the septin complex lacking the Sept6 AH domain binds to *S. flexneri* just as well as wild-type septin complexes via reconstitution methods). Some other membrane binding site might be able to contribute to this interaction, although non-specifically. As AH domains have been shown to have lipid-binding specificity, it would be interesting to test if septin complexes lacking the AH domain were mixed with *S. flexneri* lacking cardiolipin (or even non-actively dividing bacteria). One might hypothesize that these complexes (lacking the AH domain) might even bind better than wild-type complexes. Therefore, the AH domain would be providing a "specificity cue" required for septins to faithfully entrap *S. flexneri*. Even if these experiments are not performed, it would be helpful if the authors commented on this in the text.

4.) What do the septin filaments / structures (lacking the AH domain) that do entrap *S. flexneri*

look like. Are they different from those formed from wild-type complexes. Moreover, are the kinetics of septin binding to *S. flexneri* different at all (relative to AH truncation and wild-type)? If they are different, can the authors put a representative image of this in the figure relative to wild-type and if they are not, it would be helpful if the authors could comment on this in the text.

Minor comments:

- 1.) It is interesting that septins entrap the bacteria *S. flexneri*, *M. marinum*, and *M. smegmatis*. It is also important to note that these are all rod-shaped bacteria and given that septins have been shown to be sensors of cell shape, I wonder if spheroplasting the cells (to change their shape) would result similar or even different septin recruitment. This could be one way to further tease out the relative contributions between septin-cardiolipin binding vs. septin-shape recognition.
- 2.) Figure 1f: the septin rings that form around *M. marinum* and *M. smegmatis* almost appear periodic. It might be fruitful to examine the spacing between septin filaments / rings on these organisms using the Cryo-ET approach. I imagine the authors already have the data for this, and it might reveal something interesting.
- 3.) It would be helpful to the reader if the authors added the septin concentration used for a given in vitro reconstitution experiment within the figure legends. Moreover, it is unclear why the authors chose this concentration specifically (240 nM). Is this the measured septin concentration observed in HeLa cells?
- 4.) For Figure 2a and b: Is the scale bar 5 μm (as in described in the figure legend for panel E)?
- 5.) Figure 3 would benefit by putting the AH diagram generated by Heliquist (from extended data fig 4) was placed into panel A.
- 6) In places where fields of small bacteria are shown, it would be helpful to show zoomed in insets at higher magnification.

Reviewer #2 (Remarks to the Author):

In a series of publications the Mostowy group demonstrated previously that members of the septin family of cytoskeletal GTP-binding proteins form cages around intracellular bacteria such as *Shigella* or *Mycobacterium* prompting the autophagic degradation of these microbial invaders. Previous studies were conducted in cell culture as well as zebrafish models and proffered different models to explain how septin cages are formed around intracytoplasmic bacteria. The current work describes the development of an in vitro reconstitution system, in which purified recombinant septin2/6/7 complexes are incubated with different bacterial species as well as bacterial mutants to demonstrate the formation of septin cages in a cell-free system. This is an important technical advance for the field and provides a powerful system to decipher the molecular and biochemical mechanisms underlying septin cage formation and function. The presented fluorescence and cryo-EM images are certainly beautiful and the paper reads well – however, some of the central claims made in the paper are not supported by the data.

I have two main criticisms: i) the work presented here is limited insofar as it does not really provide any major novel insights that were not reported previously in studies using cell-based or zebrafish systems and, ii) some of the concepts/ interpretations put forward are not sufficiently supported by the data (- e.g. the claim that septins “recognize” IcsA seems doubtful; also, the repeated claim that the amphipathic helical (AH) domain of SEPT6 - via sensing of membrane curvature - promotes cage formation is demonstrably false based on the in vitro data shown in Fig. 3e). In order to further test whether their conclusions are justified, the authors need to incorporate proper control experiments which includes the examination of appropriate bacterial mutants (e.g. *rfaL*, *pbgA*, *icsA*/ LPS synthesis gene double mutants). Beyond conducting

experiments to (re)evaluate the fidelity of their conclusions, I believe the authors could vastly elevate the importance of their study by exploiting the power of the in vitro system to provide answers to questions that have been very difficult to address in cell-based systems, such as whether or not septin cage formation is sufficient to block actin tail formation. At the very least a revised manuscript needs to resolve the differences between the in vitro and cell-based systems that are apparent based on the differential requirement for AH.

Specific comments:

Major

- The MS claims that "an amphipathic (AH) domain encoded in SEPT6 enables septins to sense positively curved membranes and entrap bacterial cells" First of all, the authors provide no data demonstrating that the AH domain of SEPT6 (originally identified by Cannon et al (2019)) is indeed sensing membrane curvature, as demonstrated for the AH of Cdc12 by Cannon et al. Second, data in Fig. 3e show that AH is completely dispensable for septin cage formation in vitro, whereas it is required for cage formation inside cells (Fig. 3f) – these data clearly demonstrate that AH is not essential for septin cage formation. The other point to take away from these data is that the requirements for cage formation appear to be somehow different between the cell-based (requires AH) and the cell-free (doesn't require AH) systems. Why is this? The authors don't provide a model to account for their observations. This question needs to be answered to understand the potential limitations of the in vitro system. Finding conditions under which the AH is also required for cage formation in vitro (or dispensable in vivo) will likely provide the answer.
- The authors propose that "septins recognize icsA," thus at least insinuating some sort of physical interaction between the septin complex and the bacterial icsA protein. No such data are presented here. Instead the authors base their interpretation on loss- and gain-of-function genetic data – certainly a valid approach. However, for the model building the authors ignore published literature demonstrating IcsA interactions with LPS O-antigen. In other words, IcsA is likely to play an indirect role (at least in vitro) by locally displacing the O-antigen barrier to make anionic microbial lipids available for septin binding. Therefore, the authors should test whether it is the absence of O-antigen rather than the presence of IcsA which enables septin cage formation. They can do so by monitoring cage formation in icsA-deficient rough mutants (i.e. DKO icsA rfaC, icsA galU, ...). It may very well be that in the case of a rough mutant (lacking O antigen) IcsA is no longer needed for cage formation. Which also brings me to the next point
- Importantly, strains bearing mutations in the rfa genes such as rfaC express drastically truncated LPS (deep rough mutant) resulting in pleiotropic effects on bacterial cell membrane functions (see also PMID: 31273247 – which the authors may consider citing). Interpreting the data from Fig. 4 is therefore exceedingly difficult and the conclusion that LPS protects Shigella from septin cage entrapment is not sufficiently supported by the data presented here. The authors suggest that the O-antigen portion of LPS blocks septin cage formation. To test this model, the authors should expand their analysis to the galU mutant (expresses a complete lipid A + inner core) and the rfaL mutant (expresses Lipid A + inner core + outer core but no O antigen); if the authors' model is correct, then it is expected that the frequency of septin cage formation in vitro and in cells is as much increased for galU and rfaL mutants as it is for the rfaC mutant. The galU mutant is the more important mutant to test here, as it still bears the inner and outer core but lacks O-antigen.
- While the authors have established a beautiful in vitro system, they haven't used it to discover anything really novel. Their previous work had already shown a role for CL, IcsA, bacterial growth, membrane curvature, etc. in the formation of septin cages. Demonstrating direct binding of septins to the bacterial surface is novel but was certainly anticipated based on many published high resolution micrographs and the demonstration of CL-septin binding. One of the controversies in the field is whether or not septin cages actually block actin tail formation directly. This is a question that the authors could answer using this in vitro system since actin tail formation can be done in vitro.

Minor

- Whereas the icsA mutant is complemented, others are not. Why?
- Cardiolipin (CL) is present at the inner and outer bacterial membrane. Deletion of the CL biosynthesis pathway has pleiotropic effects. The authors should also test the phenotype of the CL

transporter mutant *pbgA* to determine whether specifically outer bacterial membrane CL promotes septin cage formation (see PMID: 28851846 – which should be cited)

- Please, do not refer to rough mutants as “LPS-deficient,” as it is misleading. LPS consists of the multi-acylated disaccharide lipid A, an inner core, an outer core and repeating 3-5 sugar subunits, the latter being referred to as the O antigen portion of the LPS molecule. Bacterial strains lacking the LPS heptosyltransferase 1 encoded by *rfaC* lack O-antigen, the outer core and most of the inner core but still express lipid A, the central building block of LPS. Although the resulting molecule produced by *rfaC* mutants is no longer a lipo-poly-saccharide, it is incorrect to refer to this strain as LPS-deficient, based on general convention. Rather, *rfaC* is a deep rough mutant that still produces the core structure of LPS, i.e. lipid A plus some extra sugars.
- Images throughout the manuscript would benefit from larger arrows and arrow heads
- References 7 and 23 are mixed up – e.g. line 45 should cite Goldberg et al (1993) not Robbins et al. (2001); see line 118 for the reverse
- Lines 99 – 101 “most cases ...(Fig1f)” however, Fig. 1f only shows one bacterium – I’d suggest to show representative images of fully and partially encased bacteria and provide quantification in figure
- Lines 103 – 104 “complexity of the host cell cytosol combined with limitations of resolution have prevented the visualization of septin assembly on the bacterial surface.” Ok, this made me laugh so hard. Arguably, the Mostowy lab is best known for their many beautiful papers often using high resolution microscopy visualizing septin assembly on the bacterial surface. The power of the current study, I would argue, is not allowing the visualization of septin cages (which has been done many times before) but rather the ability to take a reductionist approach to decipher mechanism and function, the area where I would hope the investigators of this study could break some new ground and present some novel biology
- It would be helpful to provide some rationale for testing T3SS and *icsB* mutants and put the negative data into some context with previously published studies
- Line 178 – also cite 17 here in addition to 10 and 27
- Lines 636/637 – second hyperinvasive is redundant

Reviewer #3 (Remarks to the Author):

Understanding cell autonomous mechanisms by which pathogens are sensed and cleared is broadly applicable and likely of general interest to the readership of Nature Communications. Here, Lobato-Márquez et. al. provide a thoughtful and mechanistic investigation into processes by which septins recognize and assemble onto bacterial pathogens. Their approach couples a newly developed in vitro reconstitution assay with cutting-edge microscopic approaches and previously described bacterial genetics to show that (1) septins recognize IcsA on the bacterial surface, (2) the AH domain of SEPT6 is required, (3) bacterial LPS restricts binding of septins to the bacterial surface, and (4) septins function in parallel with GBPs to respond to bacterial pathogens. Some concerns remain both in the interpretation of the presented data and in the depth of mechanistic insight about how septins recognize the bacterial surface, which are described below:

1. The cage around the bacteria by microscopy may appear similar once formed to that observed in cells, but the mechanism of septin filament recruitment to the bacteria may be different in vivo versus in vitro. Could the authors discuss how the septins steady state may be different or similar in the in vitro system compared to in cells. Are they similarly present as preformed complexes in cells?
2. In Figure 1, the authors show a much greater change in the amount of septins that associate with bacteria when examining Sept6 by microscopy compared to the fold change in the amount of septins that associate with bacteria by western against Sept. 7. Is this due to differences in the assays or is there a difference in the rates of Sept6 versus Sept 7 binding to the bacterial surface?
3. It is unclear whether the recruitment of septin filaments to Mycobacteria can be generalized to Shigella as the author’s show significant differences both in the percentage of mycobacteria that recruit septin and the amount of septin recruited to each bacteria. Could the author’s show that this phenotype is similar for Shigella or provide greater discussion that the mechanism of septin

recruitment between the two pathogens could be different. Also, the authors later show that Shigella IcsA is important for the recruitment, which is unlikely to be present in mycobacteria, which further suggests the mechanism are different?

4. Is the requirement of IcsA due to direct interaction of the septins with IcsA. If so, could the authors demonstrate this more clearly with a protein interaction assay?

5. Since the rfaC mutant has increased septin binding, could the authors show the impact of RfaC loss on IcsA abundance and localization?

6. It is unclear to me whether the AH domain is important for sensing the curvature of the bacterial cell or whether it is important for interaction with IcsA. It seems if it was simply curvature then there shouldn't be unipolar localization to the bacterial surface as both poles seem similarly curved.

7. Do the other septins contribute to bacterial binding or is it mediated primarily through Sept 6? Are other regions of Sept6 required for binding to the bacterial surface? Do monomers of septins bind to cells or is it only the septin complex, if monomers bind Is the binding of all of the septins dependent on IcsA or is it just Sept6?

Reviewer #4 (Remarks to the Author):

The authors establish an in vitro system for characterization of septin binding to *S. flexneri* and related bacteria. They use fluorescence microscopy, cryo-electron tomography and biochemical methods to analyze binding of recombinantly expressed and tagged SEPT2-SEPT6-SEPT8 complexes to the bacterial surface under different conditions. They identify expression of the *S. flexneri* protein IcsA and the presence of lipopolysaccharides as important factors impacting on septin filament assembly. They complement their analysis of septin binding to *S. flexneri* cells in the in vitro system by similar analyses in infected eukaryotic cells.

While I am not an expert in septin biology, the author's comprehensive structural characterization of septin in the context of pathogen immunity is to the best of my knowledge a novel and important contribution to the field that is suitable for publication in Nature Communications after the following points have been addressed:

1) Major points:

- It is good practice in the cryo-EM field to deposit reported data for public access after publication. The authors should deposit at least one example tomogram for each of the reported conditions in the Electron Microscopy Data Bank (EMDB) and include the accession codes in the paper.

- It is conceivable that the structures visible on the bacterial surfaces in Figs. 1g and 4e are Septin filaments, but the authors should also show data on control cells that have not been incubated with the recombinantly expressed septin complexes to confirm that such features are not present. In particular in Fig. 4e, the bacterial surface is covered with a whole variety of densities and it is not clear how the septin complexes were distinguished from other components that have been imaged.

- Can the authors rule out that the curvature of cells has a major impact on the molecular organization of septin assemblies and thus the distance between septin filaments and cell surface observed in Fig. 4e. In particular, the $\Delta rfaC$ cell seems much larger than the control cell. Is this a general feature? Can the authors compare the distance between WT and $\Delta rfaC$ cells of similar thickness and curvature?

2) Minor points:

- Abstract: please replace "cytosolic bacteria" by "bacteria in the cytosol of eukaryotic host cells" or something equivalent.

- The manuscript could benefit immensely from a more thorough introduction section, which seems rather superficial in this version of the manuscript. In particular, the different septin variants and their oligomerization should be introduced. Similarly, some additional details on how *S. flexneri* acts as a pathogen would be beneficial for bringing the author's findings into context.

- The authors segmented exclusively filaments that run in parallel to the viewing direction. Is there a real preferential orientation in the filament arrangement e.g. induced by compression of the cells during blotting, or is this rather due to the visibility of filaments, which certainly depends on their orientation with respect to the viewing direction? The authors should address this in the discussion section of the manuscript.

- The authors state that an amphipathic helix in SEPT6 is required for sensing positively curved membranes, but they don't show any data from their in vitro system to support this. Could the authors show some representative fluorescence microscopy images that they used for quantification of SEPT6 Δ AH binding in Fig. 3e? Naively, I would expect that SEPT6 Δ AH should not preferentially bind to the cell poles anymore, if the amphipathic helix senses positive membrane curvature.

- After deletion of the amphipathic helix in SEPT6, the authors observe a reduction in SEPT7 association with bacteria in their in vitro system, but do not comment on what would be the functional consequence. Reduced autophagy? Please address this in the discussion section of the manuscript.

- Why does the septin binding pattern in Fig. 4a change from a very localized binding to cell poles in WT cells to full entrapment of cells in Δ rfaC cells? Is this result expected? Is there an explanation? Please address this in the discussion section of the manuscript.

- I wonder how the authors can reconcile their observations and interpretations in a molecular model for septin binding to the bacterial cell surface, in particular the different septin – membrane distances observed with and without *S. flexneri* lipopolysaccharides. Which molecular components would septin bind to in the two distinct scenarios? Please elaborate on these aspects in the discussion section or - even better - add a cartoon.

- In the methods section, please provide information on the dose rate on the cameras (electrons per pixel per second) and whether the dose per tilt image was adjusted for higher tilts or kept constant.

- The authors should provide details on how the membrane – filament distances were computed in the methods section. They only state that a "homemade python script" was used. This script should also be made available according to Nature Communication's policies.

REVIEWER COMMENTS

Reviewer #1 (Remarks to the Author):

Summary: In this study, Lobato-Marquez et. al. continue their excellent work on the role of septins in innate immunity by shedding light onto the mechanisms of bacterial cage formation by the septin cytoskeleton. They developed a powerful in vitro reconstitution platform and use this tool to show bacterial factors required for septin binding and cage formation. They have also identified an amphipathic helix within the C-terminus of Sept6, which is required for septin-mediated entrapment of *S. flexneri* in infected HeLa cells, providing further mechanistic insight into how septins interact with membranes during infection. Overall, I find this manuscript to be of high interest to the field.

I had no concerns with how the majority of experiments were designed / performed, however there were several points in the manuscript where I find it important that the authors elaborate on their data. Therefore, most of my comments can be addressed by revising the text.

We thank Reviewer #1 for their enthusiasm.

“Major” comments:

1.) Regarding lines 98-102 “ Both mycobacterial species are highly recognized by septins in vitro (67.8 ± 8.1 % *M. smegmatis* and 65.2 ± 8.8 % *M. marinum*) and in most cases septins assemble into cage-like structures that cover the entire surface of mycobacterial cells (Fig. 1f). This situation contrasted with *S. flexneri*, where septin binding is dependent on bacterial growth and mainly occurs at one pole of the bacterial cell.” This is very interesting. Can the authors speculate as to why this might be the case? Might it have to do something with the respective affinity for septins and a given bacteria? It might not even be a different mechanism (as the authors propose in the discussion), as one might speculate that it takes a shorter amount of time for septins to bind and mature to fully entrap *M. smegmatis* and *M. marinum* than *S. flexneri*. It might be fruitful to perform some time-lapse imaging to watch initial septin-bacteria binding and evolution into cage formation between these organisms to make this claim. Moreover, it might be worth measuring the percentage of bound septins between these organisms?

To resolve differences between septin recruitment profiles of *S. flexneri* and *M. smegmatis*, we combined microfluidics (CellASICS ONIX2) with time-lapse fluorescence microscopy and observed real time binding events. We monitored kinetics of septin binding at the single cell level from the first observed binding event until the mean msGFP-SEPT6 intensity on the bacterial surface plateaus. While *S. flexneri* WT mostly recruits septins to one bacterial pole, *M. smegmatis* is fully covered by septins in our in vitro system (Fig. 1c and 1f). Time-lapse experiments were therefore performed using *S. flexneri* $\Delta rfaL$ where septins fully cover the bacterial surface and are also dramatically enriched at one bacterial pole (Supplementary Fig. 8a). Strikingly, septins are recruited significantly faster to the surface of *M. smegmatis* than to the surface *S. flexneri* (Fig. 5b-c). Using the time dependent increase in msGFP-SEPT6 fluorescence signal as a reporter for septin binding events, we employed a model assuming exponential kinetics and extracted binding rates (k_b) for *S. flexneri* $\Delta rfaL$ and *M. smegmatis* (Methods p17-19). From this, single-cell analysis shows that the rate of septin binding is significantly higher in *M. smegmatis* [$k_b = (20.0 \pm 0.8) \times 10^{-3} \text{ s}^{-1}$] than in *S. flexneri* $\Delta rfaL$ [$k_b = (8.71 \pm 0.5) \times 10^{-3} \text{ s}^{-1}$].

In contrast to *S. flexneri* which is a Gram-negative bacteria, mycobacteria have a cell wall with characteristics of both Gram-negative and Gram-positive bacteria (Hoffmann et al., PNAS

2008; Sani et al., PLOS Path 2010), and are covered by a mycomembrane enriched in different types of lipids (Bansal-Mutalik and Nikaido, PNAS 2014; Dulberger et al., Nat Rev Microbiol 2020). We hypothesize that differences in lipid composition of *S. flexneri* and mycobacteria are responsible for differences in septin binding. Moreover, we demonstrate that lipopolysaccharide (LPS) of *S. flexneri* WT blocks septin-membrane interactions (Supplementary Fig. 8a and 8b); considering that mycobacteria do not possess LPS, this can also explain why mycobacteria are better recognised by septins.

2.) Lines 108-110: "... septin filaments are spaced 15.9 nm from the mycobacterial outer membrane, a distance that may present the mycomembrane rich in mycobacterial lipids". This reader is unsure exactly what this means. Does this suggest that septins not in direct contact with the membrane? Is the importance of the cardiolipin to recruit some other cellular factor to the cell surface in order to bind septins? Could it potentially be lcsA? It would be helpful if the authors could clarify and further expand on this point.

The mycobacterial cell envelope (from cytosol to extracellular environment) is composed of an inner membrane, peptidoglycan, the arabinogalactan layer, the mycobacterial outer membrane (also known as mycomembrane) and an outermost capsular layer composed by proteins and lipids (including phospholipids and glycerophospholipids) (Hoffmann et al., PNAS 2008; Sani et al., PLOS Path 2010). Our data using cryoET suggests that septin filaments are not in contact with mycomembrane of *M. smegmatis* (but are distanced 15.9 nm from it, Fig. 1g and 1h) and are likely interacting with components of the capsular layer.

Considering that mycobacteria do not encode *icsA* and cardiolipin in the inner membrane (Bansal-Mutalik and Nikaido, PNAS 2014), it is unlikely that *lcsA* and cardiolipin are mediating septin binding to *M. smegmatis* or *M. marinum*.

As suggested by Reviewer #2 we designed *S. flexneri* double mutants lacking *icsA* and different components of LPS (see Reviewer #2, Point 2). These new strains clearly show that septins are binding to the bacterial surface and not *lcsA*.

3.) The authors find that by reconstituting septins lacking the Sept6 AH domain and *S. flexneri*, there was no significant difference in binding relative to wild type. However, when the AH truncation is introduced into HeLa cells, the authors observe a significant reduction in the number of septin cages entrapping *S. flexneri*. This data is very interesting and is fairly consistent with what was observed in the previous study that identified an AH within the budding yeast septin Cdc12 (Cannon et. al., 2019). In this study, septins lacking the AH domain bound to all membrane curvatures when reconstituted in vitro, however in vivo, septins lacking the AH domain failed to localize to sites of curvature. Maybe it's unsurprising that the septin complex lacking the Sept6 AH domain binds to *S. flexneri* just as well as wild-type septin complexes via reconstitution methods). Some other membrane binding site might be able to contribute to this interaction, although non-specifically. As AH domains have been shown to have lipid-binding specificity, it would be interesting to test if septin complexes lacking the AH domain were mixed with *S. flexneri* lacking cardiolipin (or even non-actively dividing bacteria). One might hypothesize that these complexes (lacking the AH domain) might even bind better than wild-type complexes. Therefore, the AH domain would be providing a "specificity cue" required for septins to faithfully entrap *S. flexneri*. Even if these experiments are not performed, it would be helpful if the authors commented on this in the text.

We thank the Reviewer for highlighting the comparison between yeast and human cells. This important comparison is included in the revised text (p5).

To investigate the role of cardiolipin in SEPT6 AH-mediated *S. flexneri*-septin interactions, we quantified (in vitro) the percentage of *S. flexneri* Δ CL (lacking cardiolipin) recruiting septins in the presence of msGFP-SEPT6WT- or msGFP-SEPT6 Δ AH-containing septin complexes. In this case, Airyscan confocal microscopy did not show a significant difference between *S. flexneri* Δ CL recruiting SEPT6WT or SEPT6 Δ AH (Rebuttal Fig. 1).

Rebuttal Fig. 1. The presence or absence of bacterial cardiolipin does not affect the binding of SEPT6WT- or SEPT6 Δ AH-containing septin complexes. Percentage of *S. flexneri* Δ CL cells recruiting septins. Data represent the mean \pm SEM of n = 1,458 (SEPT6WT) and n = 1,225 (SEPT6 Δ AH) *S. flexneri* Δ CL cells distributed in 4 independent experiments. ns, non-significant by two-tailed Student's t-test.

To test if septin concentration may influence the binding of msGFP-SEPT6WT and msGFP-SEPT6 Δ AH to *S. flexneri*, we decreased the concentration of purified septin complexes 5X (from 240 nM to 48 nM) and quantified the percentage of bacteria recruiting septins in vitro. In the case of 48 nM septins, Airyscan confocal microscopy showed a significantly decreased percentage of septin recruiting-bacteria (~60% vs ~30% for 240 nM vs 48 nM septins, respectively) but did not show significant differences in the percentage of bacteria recruiting SEPT6WT (25.0 \pm 5.0%) and SEPT6 Δ AH (34.5 \pm 10.0%) containing septin complexes.

Previous work performed in vitro used silica beads of different sizes (0.3, 1.0 and 3.0 μ m diameter) decorated with different lipid mixtures (Bridges et al., J Cell Biol 2016). Here, it was clearly shown that yeast septins can discriminate between different micron scale curvatures. However, bacteria are more complex than silica beads because: i) lipid composition of bacterial membranes is challenging to modify, ii) bacterial LPS can mask septin-membrane interactions, and iii) it is difficult to manipulate bacterial cell shape and maintain cell viability. Despite these challenges, we pharmacologically treated *S. flexneri* WT with a sublethal concentration (1 μ g/ml) of A22 and tested for septin recruitment using our in vitro reconstitution system. A22 inhibits polymerization of MreB (bacterial actin homolog), causing morphological changes to *S. flexneri* (i.e. bacterial cells shift from rod to spherical shape). Airyscan confocal microscopy showed that septins fail to bind A22 treated bacteria (data not shown). We recently showed that A22 treatment delocalizes IcsA from the bacterial pole (Krokowski et al., J Cell Sci 2019), and this delocalization is likely preventing septin recognition of *S. flexneri*. To overcome this limitation, we exploited a new LPS mutant designed to address Reviewer #2 (see Reviewer #2 points 2 and 3) called *S. flexneri* Δ rfaL (lacking the O-antigen component of LPS). Here, we treated *S. flexneri* Δ rfaL with A22 (1 μ g/ml) and incubated these bacteria with purified septins. In this case we could observe septin recruitment, but it was dramatically reduced as compared to untreated *S. flexneri* Δ rfaL (Rebuttal Fig. 2a – 2c). Airyscan confocal microscopy of in vitro caging did not show a significant difference between the percentage of *S. flexneri* cells recruiting SEPT6WT- or SEPT6 Δ AH- containing septin complexes (Rebuttal Fig. 2b and 2c). However, in our experimental conditions, the diameter of *S. flexneri* cells treated with A22 did not significantly increase as compared to untreated cells (1.36 \pm 0.02 μ m vs 1.45 \pm 0.04 μ m, respectively). We tried to further increase the diameter of bacterial cells by

using a combined treatment of sublethal concentrations of A22 (1 $\mu\text{g/ml}$) and cephalixin (8 $\mu\text{g/ml}$, an antibiotic that blocks division of bacteria) (see Rebuttal Supplementary Methods at the end of this document). In this case, the diameter of *S. flexneri* significantly increased to $1.83 \pm 0.05 \mu\text{m}$ (Rebuttal Fig. 2a), and there was increased bacterial cell lysis (data not shown). However, Airyscan confocal microscopy did not show significant differences in the percentage of *S. flexneri* cells recruiting SEPT6WT- or SEPT6 Δ AH- containing septin complexes (Rebuttal Fig. 2d and 2e). Previous work using purified yeast septins showed a significant difference in binding to beads of 1.0 μm vs 3.0 μm in diameter, and differences were even more obvious for beads of 1.0 μm vs 5.0 μm in diameter (Bridges et al., J Cell Biol 2016). In our in vitro reconstitution system, the difference in size between bacteria +/- antibiotic treatment are much smaller (e.g. 1.36 μm vs 1.83 μm , respectively). Considering this, we cannot rule out that the AH of SEPT6 plays a role in recognizing larger differences in bacterial curvature (that we are not able to test here) for septin cage entrapment in vitro.

Rebuttal Fig. 2. Small changes in the bacterial diameter do not affect the binding of SEPT6WT- or SEPT6 Δ AH-containing septin complexes. **a**, Measurement of the diameter of *S. flexneri* Δrfal cells untreated, *S. flexneri* Δrfal treated with 1 $\mu\text{g}/\text{ml}$ of A22 for 5 h, and *S. flexneri* Δrfal treated with 1 $\mu\text{g}/\text{ml}$ of A22 for 5h and 8 $\mu\text{g}/\text{ml}$ of cephalixin for 1 h. Data represents the mean \pm SEM from $n = 54$ (untreated), $n = 56$ (A22-treated) and $n = 47$ (A22 and cephalixin-treated) *Shigella* cells distributed in 3 independent experiments. ***, $p < 0.0001$ by one-way ANOVA and Tukey's post-test. **b**, Airyscan confocal images showing the binding in vitro of SEPT6WT- (top) or SEPT6 Δ AH-containing septins complexes to *S. flexneri* Δrfal cells treated with 1 $\mu\text{g}/\text{ml}$ of A22 for 5 h. Scale bar, 5 μm (inset, 2 μm). **c**, Percentage of bacteria recruiting septins in vitro. Quantifications represent mean \pm SEM from $n = 1,100$ (SEPT6WT) and $n = 835$ (SEPT6 Δ AH) *S. flexneri* Δrfal cells distributed in 3 independent experiments. ns, $p > 0.05$ by two-tailed Student's t-test. **d**, Airyscan confocal images showing the binding in vitro of

SEPT6WT- (top) or SEPT6 Δ AH-containing septins complexes to *S. flexneri* Δ rfaL cells treated with 1 μ g/ml of A22 for 5 h and 8 μ g/ml of cephalexin for 1 h. Scale bar, 5 μ m (inset, 2 μ m). **e**, Percentage of bacteria recruiting septins in vitro. Quantifications represent mean \pm SEM from $n = 614$ (SEPT6WT) and $n = 457$ (SEPT6 Δ AH) *S. flexneri* Δ rfaL cells distributed in 3 independent experiments. ns, $p > 0.05$ by two-tailed Student's t-test.

To resolve differences between septin recruitment profiles of SEPT6WT- and SEPT6 Δ AH-septin complexes to *S. flexneri*, we combined microfluidics (CellASICS ONIX2) with time-lapse fluorescence microscopy and observed real time binding events. We monitored kinetics of septin binding at the single cell level from the first observed binding event until the mean msGFP-SEPT6 intensity on the bacterial surface plateaus. Time-lapse experiments were performed using a new *S. flexneri* strain (Δ rfaL, designed in response to Reviewer #2) where septins fully cover the bacterial surface and are also dramatically enriched at one bacterial pole. Using the time dependent increase in msGFP-SEPT6 fluorescence signal as a reporter for septin binding events, we employed a model assuming exponential kinetics and extracted binding rates for *S. flexneri* Δ rfaL. Strikingly, single-cell analysis showed that SEPT6WT-containing septin complexes bind *S. flexneri* Δ rfaL cells at a significantly higher rate [$k_b = (8.71 \pm 0.5) \times 10^{-3} \text{ s}^{-1}$] than SEPT6 Δ AH-containing complexes [$k_b = (6.1 \pm 0.4) \times 10^{-3} \text{ s}^{-1}$] (Fig. 5b-d). These new data highlight the importance of the SEPT6 AH for septin complex binding to the bacterial surface in vitro.

Any differences we observe testing the SEPT6 AH in vitro (using purified septins) versus in vivo (using HeLa cells) may be a consequence of the biochemical context in which septins are found. In the cytosol of epithelial cells, different septins are well known to contribute to the septin cage, including SEPT9 and SEPT11 (Mostowy et al., Cell Host Microbe 2010). The role of the SEPT6 AH may be influenced by the presence of other septins and/or host membranes not present in our in vitro reconstitution system. We tried (unsuccessfully) to implement SEPT9 into SEPT2-SEPT6-SEPT7 hetero-oligomers in vitro (data not shown), but different labs have also shown this to be challenging (DeRose et al., Cytoskeleton 2020; Iv et al., BioRxiv, 2021). We hope the Reviewer agrees that working with septin octamers is outside the scope of this manuscript, and we comment on this future direction in the revised text (p9).

4.) What do the septin filaments / structures (lacking the AH domain) that do entrap *S. flexneri* look like. Are they different from those formed from wild-type complexes. Moreover, are the kinetics of septin binding to *S. flexneri* different at all (relative to AH truncation and wild-type)? If they are different, can the authors put a representative image of this in the figure relative to wild-type and if they are not, it would be helpful if the authors could comment on this in the text.

Airyscan confocal microscopy has not revealed any obvious differences in the quality of *S. flexneri* septin structures (in vitro, in vivo) comprised of SEPT6WT- or SEPT6 Δ AH-containing septin complexes (Supplementary Fig. 5).

To resolve the differences between septin recruitment profiles of SEPT6WT- and SEPT6 Δ AH-septin complexes to *S. flexneri*, we combined microfluidics and time-lapse fluorescence microscopy. We monitored kinetics of septin binding at the single cell level from the first observed binding event until the mean msGFP-SEPT6 intensity on the bacterial surface plateaus (see Reviewer #1 point 3). These data show that in the absence of the AH domain of SEPT6, septin complexes bind the surface of *S. flexneri* significantly slower than SEPT6WT-containing septin complexes (Fig. 5b-d).

Minor comments:

1.) It is interesting that septins entrap the bacteria *S. flexneri*, *M. marinum*, and *M. smegmatis*. It is also important to note that these are all rod-shaped bacteria and given that septins have been shown to be sensors of cell shape, I wonder if spheroplasting the cells (to change their shape) would result similar or even different septin recruitment. This could be one way to further tease out the relative contributions between septin-cardiolipin binding vs. septin-shape recognition.

Although *M. smegmatis* and *S. flexneri* are both rod-shaped bacteria, *M. smegmatis* has a smaller diameter (~0.6 μm) than *S. flexneri* (~1.0 μm).

We agree that manipulation of bacterial cell shape is of great interest. As described above (Reviewer #1, point 3), changing bacterial cell shape without altering the lipid/protein composition of membrane is challenging. Spheroplasting is harsh on bacterial cells and promotes major changes to the lipid and protein composition of bacterial cells. To address this point, we instead used a less harsh approach to disrupt bacterial cell shape (i.e. the MreB-targeting drug A22, see Reviewer #1, point 3).

2.) Figure 1f: the septin rings that form around *M. marinum* and *M. smegmatis* almost appear periodic. It might be fruitful to examine the spacing between septin filaments / rings on these organisms using the Cryo-ET approach. I imagine the authors already have the data for this, and it might reveal something interesting.

Using cryoET we measured the distance between septin filaments on *M. smegmatis* septin cages reconstituted in vitro (Fig. 1h). Under these experimental conditions, we did not observe septin rings as periodic. This information is included in the revised text (p3).

3.) It would be helpful to the reader if the authors added the septin concentration used for a given in vitro reconstitution experiment within the figure legends. Moreover, it is unclear why the authors chose this concentration specifically (240 nM). Is this the measured septin concentration observed in HeLa cells?

We used 240 nM throughout the manuscript (except for the experiments involving microfluidics; 2.4 μM). We performed a titration using 60, 120, 240 and 300 nM of purified septin complexes and observed that 240 nM was the minimum concentration required to clearly visualize septin complexes on the majority (>60%) of WT *S. flexneri*. This information is included in the revised text (Methods p15).

4.) For Figure 2a and b: Is the scale bar 5 μm (as in described in the figure legend for panel E)?

The scale bar is 5 μm . This information has been updated in the revised Fig. 2a, b.

5.) Figure 3 would benefit by putting the AH diagram generated by Heliquest (from extended data fig 4) was placed into panel A.

We update this information in the revised Fig. 3a.

6) In places where fields of small bacteria are shown, it would be helpful to show zoomed in insets at higher magnification.

Zoomed insets are now provided in Fig. 6a, Supplementary Fig. 2a, 3a, 5a and 8a.

Reviewer #2 (Remarks to the Author):

In a series of publications the Mostowy group demonstrated previously that members of the septin family of cytoskeletal GTP-binding proteins form cages around intracellular bacteria such as *Shigella* or *Mycobacterium* prompting the autophagic degradation of these microbial invaders. Previous studies were conducted in cell culture as well as zebrafish models and proffered different models to explain how septin cages are formed around intracytoplasmic bacteria. The current work describes the development of an in vitro reconstitution system, in which purified recombinant septin2/6/7 complexes are incubated with different bacterial species as well as bacterial mutants to demonstrate the formation of septin cages in a cell-free system. This is an important technical advance for the field and provides a powerful system to decipher the molecular and biochemical mechanisms underlying septin cage formation and function. The presented fluorescence and cryo-EM images are certainly beautiful and the paper reads well – however, some of the central claims made in the paper are not supported by the data.

We thank the Reviewer for describing our work as '*an important technical advance for the field*'.

We significantly revised our manuscript in agreement with all Reviewers' comments. As a result, our central claims are now more fully supported.

I have two main criticisms: i) the work presented here is limited insofar as it does not really provide any major novel insights that were not reported previously in studies using cell-based or zebrafish systems and, ii) some of the concepts/ interpretations put forward are not sufficiently supported by the data (- e.g. the claim that septins "recognize" IcsA seems doubtful; also, the repeated claim that the amphipathic helical (AH) domain of SEPT6 - via sensing of membrane curvature - promotes cage formation is demonstrably false based on the in vitro data shown in Fig. 3e). In order to further test whether their conclusions are justified, the authors need to incorporate proper control experiments which includes the examination of appropriate bacterial mutants (e.g. *rfaL*, *pbgA*, *icsA*/ LPS synthesis gene double mutants). Beyond conducting experiments to (re)evaluate the fidelity of their conclusions, I believe the authors could vastly elevate the importance of their study by exploiting the power of the in vitro system to provide answers to questions that have been very difficult to address in cell-based systems, such as whether or not septin cage formation is sufficient to block actin tail formation. At the very least a revised manuscript needs to resolve the differences between the in vitro and cell-based systems that are apparent based on the differential requirement for AH.

Major novel insights revealed in this work includes:

1- We discover that septins can directly bind bacterial surfaces in the absence of any additional host cell factors (e.g., actin, septin post-translational modification, proteins, membranes or organelles). The advantage of our in vitro reconstitution system is that we work with only 2 components: bacteria and purified septins. In contrast, using tissue culture cells, it can be difficult to precisely test how host cell factors mediate septin-bacteria interactions. As one example, it is well known that autophagy recognizes *S. flexneri* and that septins interact with the autophagy machinery (Mostowy et al., *Cell Host Microbe* 2010; Sirianni et al., *EMBO Reports* 2016; Krokowski et al., *Cell Host Microbe* 2018). In this article, we reveal that septins can directly bind the bacterial surface in the absence of autophagy machinery.

2- We reveal an unexpected role for IcsA in septin recruitment (distinct from its well-known role in actin polymerization). The new *S. flexneri* mutants we engineered (in response to

Reviewer #2) clearly demonstrate that septins bind the surface of *S. flexneri* and this interaction is promoted by IcsA-mediated disruption of bacterial LPS.

3- We discovered a new role for bacterial LPS in septin cage avoidance. Using 3 *S. flexneri* mutants targeting the synthesis of different LPS components (engineered in response to Reviewer #2), we demonstrate that the O-antigen and outer core components of the LPS protect the bacterial surface from septin recognition.

4- We visualised how septin filaments assemble on the bacterial surface using our in vitro reconstitution system and cryoET. Septins assemble into non-polar filaments and higher order structures such as bundles, rings, lattices and gauzes (Mostowy and Cossart, Nat Rev Mol Cell Biol 2012; Woods and Gladfelter, 2021 Curr Opin Cell Biol). Considering that septin filaments are 5 nm in width, their visualization in cells is highly challenging and it was unknown how septins assemble on the bacterial surface. We imaged septin cages by EM and super-resolution in the past, but failed to observe septin filaments (Mostowy et al., Cell Host Microbe 2010; Sirianni et al., EMBO Reports 2016). Our in vitro reconstitution of bacterial septin cages has shown, for the first time, that septins assemble as non-periodic filaments (and not as bundles, rings, lattices or gauzes) on the bacterial cell surface.

5- We discover that the SEPT6 AH promotes recognition of *S. flexneri* by septin complexes. New work combining our in vitro reconstitution system with microfluidics, single cell analysis and fluorescence microscopy captured septin cage formation at high temporal resolution and showed that *S. flexneri* entrapment of *S. flexneri* is slower in absence of the SEPT6 AH domain. These data are in agreement with our data in HeLa cells showing that the SEPT6 AH promotes recognition of *S. flexneri* for cage entrapment.

6- In support of our analysis performed using cryoET, we show that septins can differentially recognize *M. smegmatis* versus *S. flexneri*. To resolve the differences between septin recruitment profiles of *S. flexneri* and *M. smegmatis*, we combined microfluidics and time-lapse fluorescence microscopy. We showed that septins are recruited significantly faster to the surface of *M. smegmatis* than to the surface *S. flexneri*, and that the rate of septin binding is significantly higher for *M. smegmatis* than for *S. flexneri*. These data are consistent with our in vitro reconstitution quantifications (not performed in real time) showing that mycobacteria are significantly more entrapped in cages than *S. flexneri* WT.

7- We pioneered the approach of using bacteria as a platform to study septin assembly. Unlike other publications using supported lipid bilayers (Bridges et al., PNAS 2014; Bridges et al., J Cell Biol 2016; Szuba et al., Elife 2021), ours is the first to use bacteria. It is exciting to consider the use of different bacterial species to investigate septin biology and how it will deliver fundamental understanding of bacterial surfaces.

Considering these novel insights, we hope the Reviewer can appreciate that implementing actin tail motility is beyond the scope of this manuscript. Scientists working in the field of actin tail motility have mostly used *Listeria* (but *Listeria* are not entrapped in septin cages as shown in Mostowy et al., Cell Host Microbe 2010; Mostowy et al., J Biol Chem 2011) or *E. coli* producing IcsA as model systems (Goldberg and Theriot, PNAS 1995; Suzuki et al., EMBO J 1998). *S. flexneri* does not polymerize actin tails in vitro as efficiently as does *E. coli* producing IcsA (Magdalena and Goldberg, Cell Motil Cytoskeleton 2002). Despite these limitations, we have been trying (since 2019, interrupted by Covid) to make actin tails in vitro, but we have not been successful so far. We agree that investigating the coordination of actin polymerization and septin assembly using bacteria in vitro is an exciting research avenue, and will be the subject of future investigation by our lab.

We are grateful to the Reviewer, as her/his suggestions significantly strengthened our conclusions and led to in depth understanding of how *S. flexneri* is recognized by septins. In response to the Reviewer, we constructed the following bacterial mutants:

1- 2 new LPS mutants: *S. flexneri* $\Delta rfaL$ and $\Delta galU$, to confirm our observation that bacterial LPS protects *S. flexneri* from septin recognition (Supplementary Fig. 8a and 8b).

2- 3 new double mutants: *S. flexneri* $\Delta rfaL\Delta icsA$, $\Delta galU\Delta icsA$ and $\Delta rfaC\Delta icsA$. These new strains demonstrate that septins do not bind directly to *lcsA*, but instead (as suggested by the Reviewer) *lcsA* is likely disturbing LPS at the bacterial pole (Fig. 6a and 6b). This disruption enables septins to interact with the bacterial surface.

3- 3 new complemented strains: we complemented the lack of *rfaL*, *galU* and *rfaC* by expressing these genes in trans from 3 plasmids (Supplementary Fig. 8). In all cases, complemented strains show the same septin recruitment as *S. flexneri* WT.

Specific comments:

Major

1- The MS claims that “an amphipathic (AH) domain encoded in SEPT6 enables septins to sense positively curved membranes and entrap bacterial cells” First of all, the authors provide no data demonstrating that the AH domain of SEPT6 (originally identified by Cannon et al (2019)) is indeed sensing membrane curvature, as demonstrated for the AH of Cdc12 by Cannon et al. Second, data in Fig. 3e show that AH is completely dispensable for septin cage formation in vitro, whereas it is required for cage formation inside cells (Fig. 3f) – these data clearly demonstrate that AH is not essential for septin cage formation. The other point to take away from these data is that the requirements for cage formation appear to be somehow different between the cell-based (requires AH) and the cell-free (doesn't require AH) systems. Why is this? The authors don't provide a model to account for their observations. This question needs to be answered to understand the potential limitations of the in vitro system. Finding conditions under which the AH is also required for cage formation in vitro (or dispensable in vivo) will likely provide the answer.

In agreement with comments made by Reviewer #1 (see point 3), our data are consistent with previous observations that yeast septins lacking the AH domain bound to membrane curvature when reconstituted in vitro, but failed to localize to membrane curvature in vivo (Cannon et al., J Cell Biol 2019).

Here, we show using HeLa cells that in the absence of SEPT6 AH, septin complexes no longer localize to positive membrane curvature inside the host cell (Fig. 3d). Moreover, our in vitro reconstitution system using purified SEPT6 Δ AH-containing septin complexes showed a significant reduction in the amount of septins bound to *S. flexneri* (Fig. 3g and 3h). The AH is known to enable binding and preference for micron scale curvature (Drin and Antony, FEBS Lett 2010). As described for Reviewer #1 point3, we further explored the importance of the AH in *S. flexneri* - septin interactions and updated our revised manuscript accordingly (p5).

2- The authors propose that “septins recognize *icsA*,” thus at least insinuating some sort of physical interaction between the septin complex and the bacterial *icsA* protein. No such data are presented here. Instead the authors base their interpretation on loss- and gain-of-function genetic data – certainly a valid approach. However, for the model building the authors ignore published literature demonstrating *lcsA* interactions with LPS O-antigen. In other words, *lcsA* is likely to play an indirect role (at least in vitro) by locally displacing the O-antigen barrier to make anionic microbial lipids available for septin binding. Therefore, the authors should test whether it is the absence of O-

antigen rather than the presence of lcsA which enables septin cage formation. They can do so by monitoring cage formation in icsA-deficient rough mutants (i.e. DKO icsA rfaC, icsA galU, ...). It may very well be that in the case of a rough mutant (lacking O antigen) lcsA is no longer needed for cage formation. Which also brings me to the next point

We thank the Reviewer for their suggestions, enabling us to better understand how septins recognize *S. flexneri* and the precise role of lcsA during this process.

We tested for protein-protein interaction between septins and lcsA using pulldown assays, but these experiments failed to show a direct interaction (data not shown). We tested (using our in vitro system) double knockout mutants *S. flexneri* $\Delta rfaL\Delta icsA$, $\Delta galU\Delta icsA$ and $\Delta rfaC\Delta icsA$ (Fig. 6a and 6b). In the case of *S. flexneri* $\Delta rfaC\Delta icsA$, we clearly observed by Airyscan confocal microscopy that septins bind to bacterial surfaces in the absence of lcsA (Fig. 6a and 6b). We additionally employed bacterial sedimentation assays and western blotting to measure the amount of SEPT7 bound to $\Delta rfaC$ and $\Delta rfaC\Delta icsA$. Consistent with fluorescence microscopy data, we did not observe significant differences in the amount of SEPT7 bound to $\Delta rfaC$ and $\Delta rfaC\Delta icsA$ (Fig. 6c). In the case of *S. flexneri* $\Delta galU\Delta icsA$, we observed that septin recruitment is less homogeneous than the single $\Delta galU$ mutant, suggesting that the absence of lcsA can affect the distribution of septins on bacterial surfaces but not the overall recognition of bacterial cells by septins (Fig. 6a and 6b). In the case of *S. flexneri* $\Delta rfaL\Delta icsA$, the percentage of septin-recruiting bacteria is significantly reduced as compared to the single *S. flexneri* $\Delta rfaL$ mutant, strongly suggesting that lcsA can promote septin recruitment by forming pores in the O-antigen component and outer core of LPS (Fig. 6a and 6b). From these experiments (see also point 3) we conclude there is no direct septin-lcsA interaction, and that lcsA can disrupt LPS at the bacterial pole to enable septin-bacteria interactions. We highlight these new results in the revised text (p8).

3- Importantly, strains bearing mutations in the rfa genes such as rfaC express drastically truncated LPS (deep rough mutant) resulting in pleiotropic effects on bacterial cell membrane functions (see also PMID: 31273247 – which the authors may consider citing). Interpreting the data from Fig. 4 is therefore exceedingly difficult and the conclusion that LPS protects Shigella from septin cage entrapment is not sufficiently supported by the data presented here. The authors suggest that the O-antigen portion of LPS blocks septin cage formation. To test this model, the authors should expand their analysis to the galU mutant (expresses a complete lipid A + inner core) and the rfaL mutant (expresses Lipid A + inner core + outer core but no O antigen); if the authors' model is correct, then it is expected that the frequency of septin cage formation in vitro and in cells is as much increased for galU and rfaL mutants as it is for the rfaC mutant. The galU mutant is the more important mutant to test here, as it still bears the inner and outer core but lacks O-antigen.

To test if the O-antigen of *S. flexneri* protects the bacterial surface from septin binding, we designed 2 new *S. flexneri* LPS mutants ($\Delta rfaL$ and $\Delta galU$) and used them in our in vitro reconstitution system. These new strains showed no significant difference in the percentage of bacteria recruiting septins as compared to our original mutant $\Delta rfaC$ (Supplementary Fig. 8a and 8b) and bind septins significantly more than *S. flexneri* WT (Supplementary Fig. 8a and 8b). *S. flexneri* $\Delta galU$ showed a similar pattern of bound septins as compared to *S. flexneri* $\Delta rfaC$, i.e. the entire bacterial surface is covered by septins. In the case of *S. flexneri* $\Delta rfaL$, septins cover the entire bacterial surface and are also enriched at one bacterial pole (i.e. similar to *S. flexneri* WT). Considering that septin recruitment is decreased in the double knockout mutant *S. flexneri* $\Delta rfaL\Delta icsA$ (see Reviewer #2 point 2), we conclude that lcsA is important for bacterial recognition by septins when the O-antigen component and outer core of LPS is present. All *S. flexneri* LPS mutants recovered the WT phenotype when

corresponding deleted genes (*rfaL*, *galU*, *rfaC*) were produced in trans from a plasmid. Together, our conclusion that LPS protects *Shigella* from septin cage entrapment is strongly supported by these new data. We include this information, and also cite PMID: 31273247, in the revised text (p6-7).

4- While the authors have established a beautiful in vitro system, they haven't used it to discover anything really novel. Their previous work had already shown a role for CL, IcsA, bacterial growth, membrane curvature, etc. in the formation of septin cages. Demonstrating direct binding of septins to the bacterial surface is novel but was certainly anticipated based on many published high resolution micrographs and the demonstration of CL-septin binding. One of the controversies in the field is whether or not septin cages actually block actin tail formation directly. This is a question that the authors could answer using this in vitro system since actin tail formation can be done in vitro.

Thank you for recognising our '*beautiful in vitro system*'. Novel discovery from our in vitro system is detailed above (initial comment to Reviewer #2).

As mentioned above (initial comment to Reviewer #2), we tried implementing actin tail polymerization in our in vitro system but so far have not been successful.

Minor

- Whereas the *icsA* mutant is complemented, others are not. Why?

We complemented all mutants used in this manuscript (see Reviewer #2 point 3).

- Cardiolipin (CL) is present at the inner and outer bacterial membrane. Deletion of the CL biosynthesis pathway has pleiotropic effects. The authors should also test the phenotype of the CL transporter mutant *pbgA* to determine whether specifically outer bacterial membrane CL promotes septin cage formation (see PMID: 28851846 – which should be cited)

We tried to design a $\Delta pbgA$ mutant, however we failed to obtain any positive clones after 10 attempts. We had problems in the past mutating other genes regulating lipid transport (it took us 4 months and the use of different approaches to obtain *S. flexneri* $\Delta mlaA$, $\Delta pldA$ and $\Delta pagP$ mutant strains). Studying bacterial membrane lipid composition in the context of septin cage entrapment is of great interest, and in the future will be the subject of intense investigation. As we could not engineer this *S. flexneri* mutant we did not include *pbgA* in the revised manuscript.

- Please, do not refer to rough mutants as "LPS-deficient," as it is misleading. LPS consists of the multi-acylated disaccharide lipid A, an inner core, an outer core and repeating 3 -5 sugar subunits, the latter being referred to as the O antigen portion of the LPS molecule. Bacterial strains lacking the LPS heptosyltransferase 1 encoded by *rfaC* lack O-antigen, the outer core and most of the inner core but still express lipid A, the central building block of LPS. Although the resulting molecule produced by *rfaC* mutants is no longer a lipo-poly-saccharide, it is incorrect to refer to this strain as LPS-deficient, based on general convention. Rather, *rfaC* is a deep rough mutant that still produces the core structure of LPS, i.e. lipid A plus some extra sugars.

We agree. This is updated in the revised text.

- Images throughout the manuscript would benefit from larger arrows and arrow heads

Larger arrows and arrow heads are provided throughout our revised Figures (for example, see Fig. 1g, 3d and 4c).

- References 7 and 23 are mixed up – e.g. line 45 should cite Goldberg et al (1993) not Robbins et al. (2001); see line 118 for the reverse

These references have been updated (line 46 and 128).

- Lines 99 – 101 “most cases ...(Fig1f)” however, Fig. 1f only shows one bacterium

To complement Fig. 1f, we also include a new Supplementary Fig. 2a showing a wider field of view.

– I’d suggest to show representative images of fully and partially encased bacteria and provide quantification in figure

We now include a representative image showing septin recruitment to *M. smegmatis* and *M. marinum* (Supplemental Fig. 2a).

- Lines 103 – 104 “complexity of the host cell cytosol combined with limitations of resolution have prevented the visualization of septin assembly on the bacterial surface.” Ok, this made me laugh so hard. Arguably, the Mostowy lab is best known for their many beautiful papers often using high resolution microscopy visualizing septin assembly on the bacterial surface. The power of the current study, I would argue, is not allowing the visualization of septin cages (which has been done many times before) but rather the ability to take a reductionist approach to decipher mechanism and function, the area where I would hope the investigators of this study could break some new ground and present some novel biology

Our in vitro reconstitution system has enabled the precise visualization of septin assemblies on the bacterial surface at the nanometer scale using cryoET. As a result, we could describe for the first time that septins assemble into filaments (and not rings, bundles, gauze or lattice-like structures) on the bacterial cell surface. Super-resolution microscopy has successfully described cage like structures (Mostowy et al., Cell Host Microbe 2010; Sirianni et al., EMBO Rep 2016), but failed to provide the resolution required to visualize septin filaments on the bacterial surface. We more clearly highlight this important message in the revised text (p3, and p9).

- It would be helpful to provide some rationale for testing T3SS and icsB mutants and put the negative data into some context with previously published studies

The T3SS is crucial for host cell invasion and escape of *S. flexneri* to the cytosol (Allaoui et al., Mol Microbiol 1993; Schnupf and Sansonetti, Microbiol Spectr 2019); as a result, *S. flexneri* mutants lacking the T3SS are not entrapped in septin cages during infection (Mostowy et al., Cell Host Microbe 2010). Here, we tested whether the T3SS may have a direct role in septin caging, and show that the *S. flexneri* T3SS does not influence septin-bacteria interactions under our experimental conditions in vitro (Fig. 2b).

Inside HeLa cells, IcsB is a T3SS effector well known to block autophagy (Ogawa et al., Science 2005) and septin caging (Mostowy et al., Cell Host Microbe 2010). Here, we tested whether IcsB may have a direct role in septin caging, and show that IcsB does not influence septin-bacteria interactions under our experimental conditions in vitro (Fig. 2b).

Our rationale for testing T3SS and IcsB mutants is provided in the revised text (p4-5). We also put our negative data into the context of previously published studies (Alloui et al., Mol Microbiol 1993; Ogawa et al., Science 2005; Mostowy et al., Cell Host Microbe 2010).

- Line 178 – also cite 17 here in addition to 10 and 27

This reference has been added (line 194).

- Lines 636/637 – second hyperinvasive is redundant

This redundant text has been removed.

Reviewer #3 (Remarks to the Author):

Understanding cell autonomous mechanisms by which pathogens are sensed and cleared is broadly applicable and likely of general interest to the readership of Nature Communications. Here, Lobato-Márquez et. al. provide a thoughtful and mechanistic investigation into processes by which septins recognize and assemble onto bacterial pathogens. Their approach couples a newly developed in vitro reconstitution assay with cutting-edge microscopic approaches and previously described bacterial genetics to show that (1) septins recognize IcsA on the bacterial surface, (2) the AH domain of SEPT6 is required, (3) bacterial LPS restricts binding of septins to the bacterial surface, and (4) septins function in parallel with GBPs to respond to bacterial pathogens. Some concerns remain both in the interpretation of the presented data and in the depth of mechanistic insight about how septins recognize the bacterial surface, which are described below:

Thank you. We addressed all concerns in full.

1. The cage around the bacteria by microscopy may appear similar once formed to that observed in cells, but the mechanism of septin filament recruitment to the bacteria may be different in vivo versus in vitro. Could the authors discuss how the septins steady state may be different or similar in the in vitro system compared to in cells. Are they similarly present as preformed complexes in cells?

During purification of septin complexes, we select for SEPT2-SEPT6-SEPT7 complexes that mostly elute as hetero-hexamers. We selected SEPT2-SEPT6-SEPT7 because this septin hetero-oligomer is well characterized (Sirajuddin et al., Nature 2007; Mendonca et al., Cytoskeleton 2019) and these three septins assemble into cages in human epithelial cells upon *S. flexneri* infection (Mostowy et al., Cell Host Microbe 2010; Sirianni et al., EMBO Rep 2016; Krokowski et al., Cell Host Microbe 2018). In addition, septins are known to assemble as filaments on lipid bilayers (Tanaka-Takiguchi et al., Curr Biol 2009; Bridges et al., PNAS 2014). In our in vitro reconstitution system, the surface of bacterial cells provides a platform that promotes septin assembly into cage-like structures.

2. In Figure 1, the authors show a much greater change in the amount of septins that associate with bacteria when examining Sept6 by microscopy compared to the fold change in the amount of septins that associate with bacteria by western against Sept. 7. Is this due to differences in the assays or is there a difference in the rates of Sept6 versus Sept 7 binding to the bacterial surface?

We do not think there is a significant difference between the rates of SEPT6 versus SEPT7 binding to the bacterial surface. Septin complexes are comprised of SEPT2-SEPT6-SEPT7-SEPT7-SEPT6-SEPT2 (i.e., equal representation of SEPT6 and SEPT7). The difference

between the fluorescence data (SEPT6) and the WB data (SEPT7) is probably due to WB sample processing. In this case, during the pull down of bacteria and the septin filaments bound to them, we wash the sample 2-3 x (spinning down at low speed) to try and eliminate unbound septins (Methods, p15-16). However, there is always some residual unbound septins and thus the amount of septins bound to bacteria may appear greater by microscopy than by WB.

3. It is unclear whether the recruitment of septin filaments to Mycobacteria can be generalized to *Shigella* as the author's show significant differences both in the percentage of mycobacteria that recruit septin and the amount of septin recruited to each bacteria. Could the author's show that this phenotype is similar for *Shigella* or provide greater discussion that the mechanism of septin recruitment between the two pathogens could be different. Also, the authors later show that *Shigella* IcsA is important for the recruitment, which is unlikely to be present in mycobacteria, which further suggests the mechanism are different?

Septin recruitment to mycobacteria and *S. flexneri* is discussed in response to Reviewer #1 point 1 and Reviewer #2 point 2. Our text and Figures have been revised accordingly (p7 and p9).

4. Is the requirement of IcsA due to direct interaction of the septins with IcsA. If so, could the authors demonstrate this more clearly with a protein interaction assay?

The requirement of IcsA is addressed in response to Reviewer #2 point 2. Our manuscript has been revised accordingly (p7-8).

5. Since the *rfaC* mutant has increased septin binding, could the authors show the impact of RfaC loss on IcsA abundance and localization?

The impact of deleting genes involved in the synthesis of LPS (*rfaL*, *galU*) has previously been shown to alter the unipolar localization of IcsA (Sandlin et al., Infect Immun 1995; Sandlin et al., Mol Microbiol 1996). In *S. flexneri*, disrupting synthesis of LPS components will delocalize IcsA, yet these bacteria are still able to polymerize actin (even though actin tail motility is compromised). These data are consistent with our observation that *S. flexneri* Δ *rfaC* polymerizes fewer actin tails than *S. flexneri* WT (Supplementary Fig. 6c and 6d). We include this information in the revised text (p6).

6. It is unclear to me whether the AH domain is important for sensing the curvature of the bacterial cell or whether it is important for interaction with IcsA. It seems if it was simply curvature then there shouldn't be unipolar localization to the bacterial surface as both poles seem similarly curved.

New experiments suggest that IcsA disrupts LPS at the bacterial pole and creates pores to permit the interaction of septins with the bacterial surface (Reviewer #2, point 2). This model can explain why septins are recruited to one pole of the *S. flexneri* WT cell (i.e. where IcsA is placed). According to this model, the SEPT6 AH domain would contribute by sensing curvature at the same bacterial pole where IcsA is located (and where the bacterial surface is interacting with septins).

7. Do the other septins contribute to bacterial binding or is it mediated primarily through Sept 6? Are other regions of Sept6 required for binding to the bacterial surface? Do monomers of septins bind to cells or is it only the septin complex, if monomers bind is the binding of all of the septins dependent on IcsA or is it just Sept6?

Septins assemble as hetero-oligomeric complexes and filaments (Mostowy and Cossart, Nat Rev Mol Cell Biol 2012; Woods and Gladfelter, Curr Opin Cell Biol 2021). However, it has been suggested that different septins can play different roles (Spilliotis and McMurray, Mol Biol Cell 2020). For example, we previously demonstrated that SEPT9, but not SEPT2 or SEPT6, can bind bacterial cardiolipin (Krokowski et al., Cell Host Microbe 2018). In the case of SEPT7, it is not known if it can bind cardiolipin (because SEPT7 will aggregate when purified alone, Zhu et al., J Biol Chem, 2008).

All septins possess a polybasic domain important for binding to the hydrophilic portion of anionic phospholipids (Mostowy and Cossart, Nat Rev Mol Cell Biol 2012). However, the precise role of this domain has not yet been tested in our experimental conditions. In the future, it will be of great interest to test the precise role of septin domains (including polybasic, GTP, septin unique element) in the binding of bacterial surfaces.

Reviewer #4 (Remarks to the Author):

The authors establish an in vitro system for characterization of septin binding to *S. flexneri* and related bacteria. They use fluorescence microscopy, cryo-electron tomography and biochemical methods to analyze binding of recombinantly expressed and tagged SEPT2-SEPT6-SEPT8 complexes to the bacterial surface under different conditions. They identify expression of the *S. flexneri* protein *lcsA* and the presence of lipopolysaccharides as important factors impacting on septin filament assembly. They complement their analysis of septin binding to *S. flexneri* cells in the in vitro system by similar analyses in infected eukaryotic cells.

While I am not an expert in septin biology, the author's comprehensive structural characterization of septin in the context of pathogen immunity is to the best of my knowledge a novel and important contribution to the field that is suitable for publication in Nature Communications after the following points have been addressed:

We thank the Reviewer for comments to improve our manuscript.

1) Major points:

- It is good practice in the cryo-EM field to deposit reported data for public access after publication. The authors should deposit at least one example tomogram for each of the reported conditions in the Electron Microscopy Data Bank (EMDB) and include the accession codes in the paper.

We deposited all tomograms presented in the manuscript. Accession codes (shown below) have been included in the revised manuscript. These are:

- *M. smegmatis* + septins: #EMD-12562
- *M. smegmatis* control: #EMD-12565
- *S. flexneri* WT + septins: #EMD-12571
- *S. flexneri* WT control: #EMD-12578
- *S. flexneri* $\Delta rfaC$ + septins: #EMD-12579
- *S. flexneri* $\Delta rfaC$ control: #EMD-12580

- It is conceivable that the structures visible on the bacterial surfaces in Figs. 1g and 4e are Septin filaments, but the authors should also show data on control cells that have not been incubated with the recombinantly expressed septin complexes to confirm that such features are not present. In particular in Fig. 4e, the bacterial surface is covered with a whole variety of densities and it is not clear how the septin complexes were distinguished from other components that have been imaged.

We now include control tomograms in the revised manuscript (Supplemental Fig. 2c and 9).

- Can the authors rule out that the curvature of cells has a major impact on the molecular organization of septin assemblies and thus the distance between septin filaments and cell surface observed in Fig. 4e. In particular, the $\Delta rfaC$ cell seems much larger than the control cell. Is this a general feature? Can the authors compare the distance between WT and $\Delta rfaC$ cells of similar thickness and curvature?

The septin assemblies we visualise in our experimental conditions using *S. flexneri* WT or $\Delta rfaC$ appear to be the same (Fig. 4a, 4c and 6d). We agree that *S. flexneri* $\Delta rfaC$ mutant can have a slightly different morphology (i.e. more rounded) than *S. flexneri* WT (Xu et al., PeerJ 2016; Pagnout et al., Sci Rep 2019). To rule out that morphology of *S. flexneri* $\Delta rfaC$ may affect how septin assemblies interact with the bacterial surface, we performed cryoET and show a new tomogram where the bacterial diameter ($\sim 0.89 \mu\text{m}$) is the same as *S. flexneri* WT (tomogram shown in Fig. 4e)(Rebuttal Fig. 3).

Rebuttal Fig. 3. Correlative cryo-light microscopy and electron tomography of an in vitro reconstituted *S. flexneri* $\Delta rfaC$ cell. Scale bars, 5 μm (fluorescence microscopy) and 200 nm (cryoET and segmentation model).

Our conclusions are further supported by the treatment of *S. flexneri* WT with A22 (Rebuttal Fig. 2). In this case, bacterial cells adopt a spherical shape but septin recruitment is reduced, likely because IcsA gets delocalized from the bacterial pole (and therefore septins cannot access the bacterial surface blocked by LPS; see Reviewer #1 point 3).

2) Minor points:

- **Abstract:** please replace “cytosolic bacteria” by “bacteria in the cytosol of eukaryotic host cells” or something equivalent.

This has been replaced (line 35).

- **The manuscript could benefit immensely from a more thorough introduction section, which seems rather superficial in this version of the manuscript. In particular, the different septin variants and their oligomerization should be introduced. Similarly, some additional details on how *S. flexneri* acts as a pathogen would be beneficial for bringing the author’s findings into context.**

We extended the Introduction in our revised manuscript, providing further information on these points suggested by the Reviewer (highlighted text p2).

- The authors segmented exclusively filaments that run in parallel to the viewing direction. Is there a real preferential orientation in the filament arrangement e.g. induced by compression of the cells during blotting, or is this rather due to the visibility of filaments, which certainly depends on their orientation with respect to the viewing direction? The authors should address this in the discussion section of the manuscript.

Our work using cryoET has revealed, for the first time, that septin assemble as filaments on the bacterial cell surface at the pole. Due to limited tilting range (or “missing wedge”) and projection imaging in cryoET, it is not possible to visualize septin filaments across the entire bacterial surface. In addition, side views of corresponding tomograms (See Fig. 1g and 6d) clearly indicate that bacterial cell shape was maintained (and cells were not compressed after blotting). From these experiments using cryoET, we conclude that septin filaments on the bacterial surface represent the native assembly of septins (and are not derived from sample manipulation).

Our Discussion is updated to include this information (p9).

- The authors state that an amphipathic helix in SEPT6 is required for sensing positively curved membranes, but they don't show any data from their in vitro system to support this. Could the authors show some representative fluorescence microscopy images that they used for quantification of SEPT6 Δ AH binding in Fig. 3e? Naively, I would expect that SEPT6 Δ AH should not preferentially bind to the cell poles anymore, if the amphipathic helix senses positive membrane curvature.

The role of the SEPT6 AH domain has been addressed in response to Reviewer #1 point 3.

- After deletion of the amphipathic helix in SEPT6, the authors observe a reduction in SEPT7 association with bacteria in their in vitro system, but do not comment on what would be the functional consequence. Reduced autophagy? Please address this in the discussion section of the manuscript.

We include new comments on the functional consequence of AH domain deletion from SEPT6 in our revised Discussion (p9).

- Why does the septin binding pattern in Fig. 4a change from a very localized binding to cell poles in WT cells to full entrapment of cells in Δ rfaC cells? Is this result expected? Is there an explanation? Please address this in the discussion section of the manuscript.

In agreement with Reviewer #2's suggestions, we performed experiments using new *S. flexneri* mutant strains that significantly furthered our understanding of mechanisms regulating septin interactions with the bacterial surface during septin cage entrapment. Specifically, we created double knockout mutants in which we removed different components of the LPS (Δ rfaL, Δ galU, Δ rfaC) and lcsA (Δ lcsA), and tested them using our in vitro reconstitution system (Fig. 6a). From these new experiments, we conclude that lcsA disrupts the LPS at the pole of *S. flexneri* WT, permitting the interaction of septins with the bacterial surface (see also response to Reviewer #2, point 2 and Reviewer #3, point 6). *S. flexneri* Δ rfaC lacks the O-antigen, outer and inner cores of LPS making the bacterial surface more accessible for septin binding (and not only restricted to one pole). This is discussed on p8-9.

- I wonder how the authors can reconcile their observations and interpretations in a molecular model for septin binding to the bacterial cell surface, in particular the different septin – membrane distances observed with and without *S. flexneri*

lipopolysaccharides. Which molecular components would septin bind to in the two distinct scenarios? Please elaborate on these aspects in the discussion section or - even better – add a cartoon.

Septins are well known to interact with lipid membranes (e.g. Bridges et al., PNAS 2014). Our previous work has shown that septins can bind the anionic phospholipid cardiolipin, which is present in the outer membrane of *S. flexneri* (Krokowski et al., Cell Host Microbe 2018). Here, we show that IcsA disrupts LPS at the bacterial pole and creates pores to permit the interaction of septins with the bacterial surface (Reviewer #2, point 2). This model can explain why septins are recruited to one pole of the *S. flexneri* WT cell (i.e. where IcsA is placed). According to this model, the SEPT6 AH domain would contribute by sensing curvature at the same bacterial pole where IcsA is located (and where septins are interacting with the bacterial surface). Moreover, septins possess other domains that may permit interactions with membrane, including the polybasic domain. It is not currently known if septins can employ different domains to bind bacterial surfaces, and how this might influence septin-bacteria spacing. This information is included in the revised text (p8).

- In the methods section, please provide information on the dose rate on the cameras (electrons per pixel per second) and whether the dose per tilt image was adjusted for higher tilts or kept constant.

The tilt series is collected in a bi-directional scheme, with -10° to 60° and then -12° to -60° in 2° incremental steps. The dose rate was kept ~ 2.1 electron/ \AA^2 per tilt and the total dose of one tilt series is ~ 130 electron/ \AA^2 . We include this information in the revised Methods section (p17).

- The authors should provide details on how the membrane – filament distances were computed in the methods section. They only state that a “homemade python script” was used. This script should also be made available according to Nature Communication’s policies.

These details are provided as Supplemental File 1.

Rebuttal Supplementary Methods (see Reviewer1, point 3)

Antibiotic treatment of *S. flexneri* during in vitro reconstitution of septin cages

To measure the effect of bacterial diameter on septin recruitment we treated *S. flexneri* WT and $\Delta rfaL$ with sublethal concentrations of an MreB-disrupting drug (A22) and a filament-inducing antibiotic (cephalexin). *S. flexneri* cultures were grown 16 h in conical polypropylene tubes containing 5 ml of M9-Tris-CAA buffer at 37°C with shaking at 200 rpm. The following day, bacterial cultures were diluted in 10 ml of fresh pre-warmed M9-Tris-CAA (1: 100 v/v) in conical polypropylene tubes and cultured for 2 h at 37°C and 200 rpm. Then, bacterial cultures were incubated with DMSO (control samples) or treated with $1\ \mu\text{g/ml}$ of A22 for 5 h at 37°C and 200 rpm. After this time, bacterial cultures were either processed for septin cage reconstitution in vitro (as described in Methods p14-15) or diluted in 10 ml of fresh pre-warmed M9-Tris-CAA (1: 2 v/v) in conical polypropylene tubes and cultured for 1 h in the presence of $8\ \mu\text{g/ml}$ of cephalexin at 37°C and 200 rpm and then processed for septin cage reconstitution in vitro. After septin cage reconstitution in vitro, bacterial samples were processed as above

and imaged by Airyscan confocal microscopy. The diameter of *S. flexneri* $\Delta rfaL$ was measured using Fiji.

REVIEWERS' COMMENTS

Reviewer #1 (Remarks to the Author):

The authors responded to my concerns and I support publication of this exciting study.

Reviewer #2 (Remarks to the Author):

This is a vastly improved revised manuscript that adequately addressed all my concerns. The authors added substantial additional data that resolved all the concerns and, I believe, helped the authors establish a much more defined and more accurate mechanistic model for the direct recognition of *Shigella* by the septin complex. This study makes important contributions to the fields of innate immunity and septin biology. Congratulations to all authors for putting together such an excellent study.

Reviewer #3 (Remarks to the Author):

My concerns have been adequately addressed by the authors.

A couple of minor comments below:

1. Line 112- should be Fig. S2C?

2. In Fig. 5, the authors suggest that the lipid composition is altered between *S. flexneri* and *M. Smegmatis*. The bacteria appear to be significantly different in size as well. Could the authors discuss what degree of membrane curvature the septins bind to and whether the shape of *Mycobacteria* or *Shigella* are better matched for septin binding.

3. Could the authors speculate more about their observation in Fig. 1 that non growing bacteria do not recruit septins in light of the observations on IcsA and LPS in Figure. 6.

Reviewer #4 (Remarks to the Author):

The authors have addressed all of my concerns and significantly strengthened their manuscript with the additional experiments they performed. It should be published in *Nature Communications*.

Reviewer #1 (Remarks to the Author):

The authors responded to my concerns and I support publication of this exciting study.

We thank the reviewer for their comments and support.

Reviewer #2 (Remarks to the Author):

This is a vastly improved revised manuscript that adequately addressed all my concerns. The authors added substantial additional data that resolved all the concerns and, I believe, helped the authors establish a much more defined and more accurate mechanistic model for the direct recognition of *Shigella* by the septin complex. This study makes important contributions to the fields of innate immunity and septin biology. Congratulations to all authors for putting together such an excellent study.

We thank the reviewer for their enthusiasm.

Reviewer #3 (Remarks to the Author):

My concerns have been adequately addressed by the authors.

A couple of minor comments below:

1. Line 112- should be Fig. S2C?

Yes, this has been updated.

2. In Fig. 5, the authors suggest that the lipid composition is altered between *S. flexneri* and *M. Smegmatis*. The bacteria appear to be significantly different in size as well. Could the authors discuss what degree of membrane curvature the septins bind to and whether the shape of *Mycobacteria* or *Shigella* are better matched for septin binding.

Our data suggest *mycobacteria* and *Shigella* both have a shape suitably matched for septin binding. Although *M. smegmatis* does have a smaller diameter (~0.6 μm) than *S. flexneri* (WT ~1.0 μm or Δrfal ~1.4 μm), work in our previous rebuttal letter (using drugs to manipulate bacterial cell shape) indicate that this level of difference in bacterial diameter does not affect septin recruitment as significantly as does lipid composition (see Reviewer1, point 3). These observations are consistent with previous reports using used silica beads of more dramatically different sizes (0.3, 1.0, 3.0 and 5.0 μm ; shape differences we were unable to test using bacterial cells) showing that septins have a preference for binding positively curved membranes of 1.0 μm in diameter, (Bridges et al., J Cell Biol 2016; Krokowski et al., Cell Host Microbe 2018).

Considering that the membrane composition of *mycobacteria* is well known to be different than that of *Shigella* (Bansal-Mutalik and Nikaido, PNAS 2014; Dulberger et al., Nat Rev Microbiol 2020), we suggest that lipid composition plays a more important role than bacterial cell shape in the recognition of these two bacterial species by septins (line 315-317 of revised manuscript).

3. Could the authors speculate more about their observation in Fig. 1 that non growing bacteria do not recruit septins in light of the observations on lcsA and LPS in Figure. 6.

The production of lcsA in *S. flexneri* is strictly dependent on bacterial growth phase (Robbins et al., Mol Microbiol 2001). We speculate that in the absence of bacterial growth, lcsA levels are decreased such that *S. flexneri* LPS is not disturbed and therefore septins cannot interact with the bacterial surface.

Reviewer #4 (Remarks to the Author):

The authors have addressed all of my concerns and significantly strengthened their manuscript with the additional experiments they performed. It should be published in Nature Communications.

We thank the reviewer for their comments and support.